# Joint Learning in the Gaussian Single Index Model

**Loucas Pillaud-Vivien** [1]   **Adrien Schertzer** [2]

## Abstract

We consider the problem of jointly learning a one-dimensional projection and a univariate function in high-dimensional Gaussian models. Specifically, we study predictors of the form $f(x) = \varphi^\star(\langle w^\star, x \rangle)$, where both the direction $w^\star \in \mathcal{S}_{d-1}$, the sphere of $\mathbb{R}^d$, and the function $\varphi^\star : \mathbb{R} \to \mathbb{R}$ are learned from Gaussian data. This setting captures a fundamental non-convex problem at the intersection of representation learning and nonlinear regression. We analyze the gradient flow dynamics of a natural alternating scheme and prove convergence, with a rate controlled by the information exponent reflecting the *Gaussian regularity* of the function $\varphi^\star$. Strikingly, our analysis shows that convergence still occurs even when the initial direction is negatively correlated with the target. On the practical side, we demonstrate that such joint learning can be effectively implemented using a Reproducing Kernel Hilbert Space (RKHS) adapted to the structure of the problem, enabling efficient and flexible estimation of the univariate function. Our results offer both theoretical insight and practical methodology for learning low-dimensional structure in high-dimensional settings.

## 1. Introduction

The problem of learning predictive functions from high-dimensional data lies at the heart of modern machine learning (Bach, 2024). Among the vast landscape of models, *single-index* structures, wherein the target depends on the data only through a low-dimensional projection, have long captured both theoretical and practical attention (Dudeja & Hsu, 2018; Ben Arous et al., 2021). In their most canonical

[1]CERMICS, CNRS, ENPC, Institut Polytechnique de Paris, Marne-la-Vallée, France [2]UMPA, ENS Lyon, 46 allée d'Italie, 69007 Lyon, France. Correspondence to: Adrien Schertzer <adrien.schertzer@ens-lyon.fr>, Loucas Pillaud-Vivien <loucas.pillaud-vivien@enpc.fr>.

*Proceedings of the 43rd International Conference on Machine Learning*, Seoul, South Korea. PMLR 306, 2026. Copyright 2026 by the author(s).

form, these models take the shape

$$x \mapsto \varphi^\star(\langle w^\star, x \rangle),$$

where $x \in \mathbb{R}^d$ denotes a high-dimensional input, $w^\star \in \mathcal{S}_{d-1}$ is an unknown direction, and $\varphi^\star : \mathbb{R} \to \mathbb{R}$ is an unknown univariate function (Kalai & Sastry, 2009; Shalev-Shwartz et al., 2010; Kakade et al., 2011). This seemingly innocuous formulation conceals a rich interplay between geometry, approximation, and statistics (Saad & Solla, 1995; Ben Arous et al., 2022; Veiga et al., 2022).

The resurgence of interest in such models (Frei et al., 2020; Zweig et al., 2023; Yehudai & Shamir, 2020; Wu, 2022) is fueled in part by their capacity to mirror the behavior of neural networks, especially in regimes where both the feature representations and the predictor are learned jointly. Indeed, when the function $\varphi^\star$ is parameterized flexibly, e.g., as an element of an infinite-dimensional space such as $L_\gamma^2(\mathbb{R})$ with respect to the Gaussian measure. The model transcends classical parametric confines and enters a semi-nonparametric realm that better captures the inductive biases of overparameterized networks (Berthier et al., 2024).

In this work, we revisit single-index models through the lens of gradient flow. Our perspective is both analytic and algorithmic: we study the continuous-time dynamics induced by gradient descent when optimizing jointly over the direction $w$ and the profile $\varphi$. To tame the infinite-dimensional nature of the problem, we exploit the Hermite basis, which offers a natural orthonormal expansion for functions in $L_\gamma^2(\mathbb{R})$. The resulting dynamics admit a concise spectral description, wherein the evolution of each Hermite coefficient unfolds in a manner coupled to the geometric progression of the feature vector $w$.

Our contributions are threefold. First, we establish convergence guarantees for this joint gradient flow, characterizing its behavior via an *information exponent* (Ben Arous et al., 2022; Damian et al., 2024) that governs the amplification of informative components. In high dimension, the effect of random initialization becomes tractable: the initial alignment between $w$ and the true direction $w^\star$ is of order $1/\sqrt{d}$, yet suffices to seed consistent recovery. Second, we dissect the dynamical coupling between direction learning and profile estimation, unveiling subtle mechanisms by which the two processes interact, sometimes harmoniously, sometimes antagonistically. Third, we introduce a practical

implementation of this methodology using a reproducing kernel Hilbert space constructed from truncated Hermite expansions (Follain & Bach, 2024). This allows us to interpolate between theoretical idealizations and algorithms amenable to numerical study.

Beyond the theoretical exposition, we corroborate our findings through a series of experiments, shedding light on the practical feasibility of this approach and validating our analytical predictions. Taken together, our results illuminate a path toward a principled understanding of joint learning in high dimensions, with ramifications for both the theory of neural networks and the design of new learning algorithms.

**Structure of the paper.** In Section 2, we formalize the problem setting, introduce the necessary mathematical tools and derive the loss in term of the summary statistics (Lemma 2.1). In Section 3, we derive the joint gradient flow equations on the summary statistics and in Section 4 present the main convergence result on the joint learning problem (Theorem 4.2). For the sake of comparison, we describe thoroughly the planted case in Proposition 4.1. Section 5 discusses the RKHS implementation and provides experimental validation. Finally, Section 7 concludes with a discussion of implications and future directions.

## 2. Problem Setup

We consider a high-dimensional learning problem in which the goal is to recover a predictive function from observations: i.e. random variables $(X, Y)$, where $X \in \mathbb{R}^d$ are high dimensional inputs and $Y \in \mathbb{R}$ is a scalar response. The data follow a single-index model (Soltanolkotabi, 2017) of the form

$$Y = \varphi^\star(\langle w^\star, X \rangle) + \varepsilon, \tag{1}$$

where: $X \sim \mathcal{N}(0, I_d)$ is a standard Gaussian vector in $\mathbb{R}^d$, $w^\star \in \mathcal{S}_{d-1} \subset \mathbb{R}^d$ is an unknown unit-norm direction (the *index vector*), $\varphi^\star \colon \mathbb{R} \to \mathbb{R}$ is an unknown measurable function (the *link function*), and $\varepsilon$ is a zero-mean noise term, independent of $X$, with finite variance.

### 2.1. Comparison with previous work.

This setting defines a classical *single-index model* (Shalev-Shwartz et al., 2010; Dudeja & Hsu, 2018), in which the response depends on the high-dimensional input only through the projection $\langle w^\star, x \rangle$. Such models are of particular interest in high-dimensional statistics and machine learning, as they offer a structured form of dimensionality reduction while retaining modeling flexibility via the nonparametric function $\varphi^\star$. For more details on these models, we refer to the recent review (Bruna & Hsu, 2025), and to (Bietti et al., 2023; Dandi et al., 2024; Abbe et al., 2023) for multivariate extensions of these.

Most prior work on single-index models has focused on settings where either the direction $w^\star$ or the function $\varphi^\star$ is known in advance, or where one of the two components is learned while the other is fixed (Kalai & Sastry, 2009; Shalev-Shwartz et al., 2010; Kakade et al., 2011; Dudeja & Hsu, 2018; Ben Arous et al., 2021). In contrast, our work addresses the more challenging scenario of *jointly* learning both $w^\star$ and $\varphi^\star$ from data. Some works (Abbe et al., 2022; Berthier et al., 2024; Dandi et al., 2024; Bietti et al., 2023) address this joint learning problem, but without training both layers simultaneously via gradient-based methods. Our analysis of the gradient flow dynamics in this joint setting provides new insights into the interplay between direction learning and function estimation, particularly in high-dimensional regimes.

### 2.2. Mathematical setup

**Function space.** To reflect this flexibility, we assume that $\varphi^\star$ belongs to the Hilbert space $L^2_\gamma(\mathbb{R})$, where $\gamma$ denotes the standard Gaussian measure on $\mathbb{R}$, and the inner product is given by

$$\langle f, g \rangle_{L^2_\gamma} := \int_{\mathbb{R}} f(z) g(z) \, d\gamma(z),$$

where $d\gamma(z) = \frac{1}{\sqrt{2\pi}} e^{-z^2/2} dz$. This choice is natural given that the latent variable $\langle w^\star, X \rangle$ is itself standard Gaussian due to the isotropy of $X \in \mathbb{R}^d$ and the unit norm of $w^\star$. It will also be useful to denote by $\mathcal{H}^j_\gamma$, the weighted Sobolev spaces $\mathcal{H}^j_\gamma := \{f \in L^2_\gamma, \text{ s.t. } f^{(j)} \in L^2_\gamma\}$, where $j \in \mathbb{N}$ and $f^{(j)}$ denote the $j$-the derivative of $f$. Obviously, this forms a nested structure of functional spaces: $L^2_\gamma = \mathcal{H}^0_\gamma \supset \mathcal{H}^1_\gamma \supset \mathcal{H}^2_\gamma \supset \ldots$ of increasing regularity.

**Model class.** The learner has access to a class of predictors parameterized by a unit vector $w \in \mathcal{S}_{d-1}$ and a function $f \in L^2_\gamma(\mathbb{R})$. The model thus takes the form:

$$\mathcal{F} := \left\{ f(\langle w, \cdot \rangle) \,|\, \text{ for } w \in \mathcal{S}_{d-1} \text{ and } f \in L^2_\gamma(\mathbb{R}) \right\},$$

which defines a semi-nonparametric family of predictors, combining a low-dimensional projection with an infinite-dimensional function class. The central object of study is the learning dynamics over this composite space.

**Joint optimization.** Our goal is to recover or approximate $(w^\star, \varphi^\star)$ by minimizing the expected squared loss

$$\mathcal{L}(f, w) := \frac{1}{2} \mathbb{E} \left[ (f(\langle w, X \rangle) - Y)^2 \right], \tag{2}$$

over $f \in L^2_\gamma(\mathbb{R})$ and $w \in \mathcal{S}_{d-1}$. In this work, we focus on the gradient flow associated with $\mathcal{L}$, that is, the continuous-time limit of gradient descent updates over both $w$ and $f$.

**Hermite basis and structure of the loss.** A crucial role in our analysis is played by the *Hermite polynomials* $(h_k)_{k \geq 0}$, which form a complete orthonormal basis of $L^2_\gamma(\mathbb{R})$, i.e. $\langle h_k, h_{k'} \rangle_{L^2_\gamma} = \delta_{kk'}$. For the sake of concreteness, we recall that for all $z \in \mathbb{R}$ $h_0(z) = 1$, $h_1(z) = z$, $h_2(z) = \frac{1}{2}(z^2 - 1) \cdots$, and that $h_k$ is even (resp. odd) when $k$ is (resp. odd) ([Bogachev, 1998](#))(Chapter 1.3). Every square-integrable function $f \in L^2_\gamma(\mathbb{R})$ thus admits a unique expansion:

$$f(z) = \sum_{k=0}^{\infty} a_k h_k(z), \text{ with } \sum_{k=0}^{\infty} a_k^2 = \|f\|^2_{L^2_\gamma} < \infty .$$

This decomposition induces a natural coordinate system for analyzing gradient flows over $f$, as each Hermite component evolves in time in a coupled yet interpretable manner with the parameters $w$. Moreover, we can represent the spaces $\mathcal{H}^k_\gamma$ with respect to summability assumption of the Hermite coefficients as $f \in \mathcal{H}^j_\gamma \Leftrightarrow \sum_{k \geq 0} k^{2j} |a_k|^2 < \infty$.

**Lemma 2.1** (Loss expansion ([Dudeja & Hsu, 2018](#)))**.** *The loss has the following expanded form*

$$\mathcal{L}(f, w) = \frac{1}{2} \|f\|^2_{L^2_\gamma} + \frac{1}{2} \|\varphi^\star\|^2_{L^2_\gamma}$$
$$- \sum_{k \geq 0} a_k a_k^\star \langle w, w^\star \rangle^k + \frac{1}{2} \text{Var}(\varepsilon) , \quad (3)$$

*That is to say that the loss rewrites* $\mathcal{L}(f, w) = \ell(a, m)$, *where*

$$\ell(a, m) = \frac{1}{2} \sum_{k \geq 0} |a_k|^2 - \sum_{k \geq 0} a_k a_k^\star \langle w, w^\star \rangle^k + C^\star_\varepsilon , \quad (4)$$

*where* $(a_k)_{k \in \mathbb{N}}, (a_k^\star)_{k \in \mathbb{N}}$ *are respectively the Hermite coefficients of* $f$ *and* $\varphi^\star$, $m = \langle w, w^\star \rangle \in [-1, 1]$ *is the correlation and* $C^\star_\varepsilon$ *is a constant that does not depend on the model* $\mathcal{F}$.

In the remaining of the article, for technical reasons, we shall assume that $\varphi^* \in \mathcal{H}^1$. This assumption ensures that the gradient flow is well defined and provides a time-uniform control on the gradient.

**High-dimensional scaling.** We are particularly interested in the high-dimensional regime where $d \gg 1$. In this setting, the inner product $\langle w, w^\star \rangle$ between the true direction and a randomly initialized unit vector is typically of order $1/\sqrt{d}$, which induces a weak signal at initialization. Understanding how such weak alignment is amplified through the learning dynamics is central to our analysis.

**Learning objective.** The overarching question we seek to answer is the following:

*Under what conditions, and at what rates, does the joint gradient flow over* $(w, f)$ *recover the structure of the planted model* (1)?

This entails both statistical and dynamical aspects: the identifiability of the target pair $(w^\star, \varphi^\star)$, and the behavior of the learning trajectory in the product space $\mathcal{S}_{d-1} \times L^2_\gamma$. In the sections that follow, we derive the gradient flow equations governing the evolution of $(w_t, f_t)$, analyze their convergence properties, and explore their practical realization via Hermite-based kernel methods.

## 3. Gradient Flow Dynamics

### 3.1. The gradient flow equations

**Gradient flow on** $L^2_\gamma \times \mathcal{S}_{d-1}$**.** In this section, we derive the equations of motions of the gradient flow on $\mathcal{L}(f, w)$. Hence, (at least formally for now), we say that $(f_t, w_t)_{t \geq 0}$ follow the gradient flow if they are solutions of the infinite dimensional system of ODEs

$$\frac{\mathrm{d}}{\mathrm{d}t} f_t = -\nabla_f^{L^2_\gamma} \mathcal{L}(f_t, w_t) , \quad (5)$$

$$\frac{\mathrm{d}}{\mathrm{d}t} w_t = -\nabla_w^{\mathcal{S}_{d-1}} \mathcal{L}(f_t, w_t) . \quad (6)$$

Let us explain the notation: for the parametric part, in order to leverage the rotation invariance of the Gaussian, we use the spherical gradient $\nabla_w^{\mathcal{S}_{d-1}} u(w) = \nabla_w u(w) - \langle w, \nabla_w u(w) \rangle w$ for any differentiable function $u$. For the non-parametric part, we use the Hilbert property of $(L^2_\gamma, \langle \cdot, \cdot \rangle_{L^2_\gamma})$ and Riesz theorem to represent the Fréchet differential of $\mathcal{L}(\cdot)$ in terms of a gradient, i.e. for all $g \in L^2_\gamma$,

$$\left\langle \nabla_f^{L^2_\gamma} \mathcal{L}(f, w), g \right\rangle_{L^2_\gamma} = \lim_{h \to 0} \frac{\mathcal{L}(f + hg, w) - \mathcal{L}(f, w)}{h} .$$
$$(7)$$

**Dynamics on** $\ell_2(\mathbb{N}) \times [-1, 1]$**.** Now that the gradient flow is (at least at the formal level) well-defined, we show that this flow on $(f_t, w_t)_{t \geq 0} \in L^2_\gamma \times \mathcal{S}_{d-1}$ induces equations of movement on the reduced parameters $((a_{k,t})_{k \in \mathbb{N}}, m_t)_{t \geq 0} \in \ell_2(\mathbb{N}) \times [-1, 1]$. In fact, the gradient structure on $f$ in the space $L^2_\gamma$ reduces to the Hilbert gradient structure on $\ell_2(\mathbb{N})$ for the coefficient $(a_k)_{k \in \mathbb{N}}$ of $f$. This corresponds to the infinite system of ODEs stated in the following Lemma.

**Lemma 3.1.** *Let* $(f_t, w_t)_{t \geq 0}$ *follow the gradient flow Eqs.*(5)-(6)*, then*

$$\frac{\mathrm{d}}{\mathrm{d}t} a_{k,t} = a_k^\star m_t^k - a_{k,t} , \text{ for all } k \in \mathbb{N} , \quad (8)$$

$$\frac{\mathrm{d}}{\mathrm{d}t} m_t = (1 - m_t^2) \sum_{k=1}^{\infty} k a_{k,t} a_k^\star m_t^{k-1} , \quad (9)$$

*where* $m_t = \langle w_t, w^\star \rangle$ *and* $f_t := \sum_{k \geq 0} a_{k,t} h_k$ .

By Fact 1 and the fact that $\varphi^* \in \mathcal{H}^1$, the reduced system is locally well posed by Cauchy–Lipschitz. The a priori estimates of Fact 1 ensure that no finite-time blow-up occurs, hence the solution is global.

## 3.2. Initialization and information exponent

We use a non-informed initialization to the problem. For the parametric part, we initialize according to the uniform distribution on the sphere. A striking consequence of this is that the correlation between initialization $w_0$ and the true direction $w^\star$ is of order $1/\sqrt{d}$.

**Lemma 3.2.** *Let* $0 < a < b$*, then*

$$\mathbb{P}(\sqrt{d}|\langle w_0, w^\star \rangle| \in [a,b]) \geq 1 - a - \frac{2e^{-b^2/4}}{b\sqrt{\pi}} - 4e^{-d/16}.$$

The proof is deferred to the appendix. Hence, when running the algorithm in practice, it is initialized with a high probability near the equator of $\mathcal{S}_{d-1}$, or at least in a band of typical size $1/\sqrt{d}$. Let

$$s = \inf\{k \in \mathbb{N}^* \,|\, a_k^* \neq 0\} \tag{10}$$

be the *information exponent* (Ben Arous et al., 2022) of $\varphi^*$. The value of this integer crucially controls the behavior of the dynamics, as we will see in the main result. Indeed, we see from Eq.(9), that at initialization the movement of $m$ is governed by the ODE $\dot{m}_t \sim a_{s,t} m_t^{s-1}$, whose typical time to escape a neighborhood of the origin depends highly on how large the exponent $s$ is. For the non-parametric part, we use an initialization on the coefficient in the Hermite basis directly and initialize such that $k^2 a_{k,0} \sim \mathcal{U}([-1,1])$. Note that thus we have $f_0 = \sum_{k \geq 0} a_{k,0} h_k \in \mathcal{H}^2$ almost surely.

**Comparison with the planted model.** In the idealized *planted model* setting, where $\varphi^\star$ is known in advance (e.g., $f_t = \varphi^\star$ or equivalently $a_{k,t} = a_k^*$ held fixed), Equation (9) becomes a closed-form ODE for $m_t$. In this case, it can be shown that: (i) if $m(0) > 0$, the alignment is reinforced and $m_t \to 1$, (ii) if $m(0) < 0$, in some cases, the gradient flow converges to the origin or to the first negative zero of the power series $x \to \sum_{k=1}^{\infty} k(a_k^\star)^2 x^{k-1}$ and in either case $w$ fails to recover $w^\star$. We highlight this result in Proposition 4.1. This stark asymmetry illustrates a surprising and important fact: *in the planted model, negative initial correlation cannot be corrected by the dynamics*. The flow becomes trapped at a fixed point, even though the loss function has a global minimum at $m = 1$ (Note also that, if $\varphi^\star$ is even, the symmetric point $m = -1$ is also a minimum). In the joint learning setting, where $f$ is learned simultaneously, the dynamics are more subtle and can escape such traps, this is a key motivation for the joint approach.

**Summary.** In summary, the gradient flow system evolves over the product space $\mathcal{S}_{d-1} \times \ell^2(\mathbb{N})$, with dynamics governed by the coupled ODEs given in Lemma 3.1. These equations encode a feedback loop: the alignment $m_t$ controls the effective signal in the functional update, while the function $f_t$, in turn, modulates the reinforcement of alignment through the coefficients $a_{k,t}$. The initialization of the

correlation, of order $1/\sqrt{d}$, structures the dynamics near the origin, where the time scales are expected to be controlled by the information exponent $s$ (see definition (10)). The precise resolution of this interaction is the subject of the next section, where we analyze the convergence behavior.

# 4. Convergence Behavior of the Joint Gradient Flow

Understanding the asymptotic behavior of the gradient flow dynamics, especially in the high-dimensional setting, is central to this work. We begin by recalling known results on the so-called *planted model*, in which the target function $\varphi^\star \in L_\gamma^2$ is fixed and known in advance, and the gradient flow acts only on the feature direction $w \in \mathcal{S}_{d-1}$. This model serves as an instructive warm-up, showing that when the initialization $m_0 := \langle w_0, w^\star \rangle$ is negatively correlated, the system fails to recover the signal.

## 4.1. The Planted Model.

In this case, the dynamics reduce to a one-dimensional ordinary differential equation for the correlation $m_t := \langle w_t, w^\star \rangle$, as previously derived:

$$\dot{m}_t = (1 - m_t^2) \sum_{k=1}^{\infty} k(a_k^\star)^2 m_t^{k-1},$$

where $a_k^\star$ are the Hermite coefficients of the function $\varphi^\star$. Let $\Phi(x) := \sum_{k=1}^{\infty} k(a_k^\star)^2 x^{k-1}$, and $\mathcal{Z}_\Phi$ the set of zeros of $\Phi$ on $[-1,1]$. Then, it is clear that $\mathcal{Z}_\Phi \cap \mathbb{R}_{>0} = \emptyset$. When $\mathcal{Z}_\Phi \cap \mathbb{R}_{<0} \neq \emptyset$, we define $\bar{m} = \min\{|z| \,|\, z \in \mathcal{Z}_\Phi \cap \mathbb{R}_{<0}\}$, and if $\mathcal{Z}_\Phi \cap \mathbb{R}_{<0} = \emptyset$, we set $\bar{m} = +\infty$ and then we have the following proposition.

**Proposition 4.1** (Bifurcations in the planted model). *Let* $m_t$ *evolve under the planted gradient flow with fixed target* $\varphi^\star \in L_\gamma^2$*, and set* $m_0 := \langle w_0, w^\star \rangle$*. Then we have*

- *If* $m_0 > 0$*, the trajectory aligns:* $\lim_{t \to \infty} m_t = 1$*.*

- *If* $m_0 < 0$*, then*

  - *if* $s \geq 3$ *is odd:* $\lim_{t \to \infty} m_t = 0$*;*
  - *if* $s$ *is even, then*

$$\lim_{t \to \infty} m_t = \begin{cases} -1, & \text{if } \bar{m} > 1, \\ -\bar{m}, & \text{if } \bar{m} < 1. \end{cases}$$

This behavior is folklore in the literature (Ben Arous et al., 2022; Bietti et al., 2023; Damian et al., 2023; Zweig & Bruna, 2023) and the result could be enhanced by proving convergence rate, but we decide not to state them here, as the only purpose of this proposition is to show that the

system could lead to learning failure in the case $m_0 < 0$. This bifurcation highlights a fundamental limitation of the planted gradient flow: although the global minimum of the loss is achieved at $m = 1$, in some cases, the dynamics cannot escape the spurious basin of attraction when the initial correlation is negative. In contrast, as we shall now demonstrate, the joint learning dynamics over both $w$ and $f$ exhibit a markedly different behavior, one which allows the system to overcome this initialization asymmetry.

### 4.2. Convergence in the Joint Learning Setting

We now turn to the full gradient flow over the product space $\mathcal{S}_{d-1} \times L^2_\gamma$, where both the direction $w_t$ and the profile function $f_t$ evolve jointly. As we have seen, the dynamics can be decomposed spectrally in the Hermite basis, with coefficients $a_{k,t} := \langle f_t, h_k \rangle$, and the correlation $m_t := \langle w_t, w^\star \rangle$. The coupled evolution equations are given in Eqs.(8)-(9). These dynamics reveal a feedback mechanism: the function $f_t$ becomes gradually aligned with the true target $\varphi^\star$, but only to the extent that the correlation $m_t$ allows it to see the correct Hermite modes. In turn, the recovery of the direction $w_t$ depends on the functional alignment via the second equation. Remarkably, this feedback loop can break the symmetry obstruction observed in the planted case. The next two theorems provide a characterization of the convergence behavior in the joint learning setting, depending on the regularity exponent $s$ of the target function $\varphi^\star$.

**Theorem 4.2.** *Let $(m, a)$ follows equations Eqs.(8)-(9). Then, with overwhelming probability on the initialization, taken as described in the previous section, there exists constants $c, C > 0$ so that,*

***Case $s \geq 2$.*** *After time $t \geq \tau_c \geq Cd^{s-1}$,*

- *if $m_0 > 0$,* $\quad |1 - m_t| \leq Ce^{-c(t-\tau_c)}$,

$$|a_{k,t} - a_k^*| \leq Cb_k e^{-c(t-\tau_c)} \text{, for all } k \in \mathbb{N} \text{,}$$

- *if $m_0 < 0$,* $\quad |1 + m_t| \leq Ce^{-c(t-\tau_c)}$,

$$|a_{k,t} - (-1)^k a_k^*| \leq Cb_k e^{-c(t-\tau_c)} \text{, for all } k \in \mathbb{N} \text{,}$$

***Case $s = 1$.*** *After time $t \geq 0$,*

- *if $a_1^* a_{1,0} > 0$,* $\quad |1 - m_t| \leq Ce^{-ct}$,

$$|a_{k,t} - a_k^*| \leq Cb_k e^{-ct} \text{, for all } k \in \mathbb{N} \text{,}$$

- *if $a_1^* a_{1,0} < 0$,* $\quad |1 + m_t| \leq Ce^{-ct}$,

$$|a_{k,t} - (-1)^k a_k^*| \leq Cb_k e^{-ct} \text{, for all } k \in \mathbb{N} \text{,}$$

*where, $b \geq 0$ is a normalized sequence of $\ell_2(\mathbb{N})$, i.e. $\|b\|_{\ell_2(\mathbb{N})} = 1$.*

Theorem 4.2 offers a precise characterization of the long-time behavior of the joint gradient flow in the Hermite basis. The evolving state of the system is described by the pair

$(m_t, a_t)$, where $m_t := \langle w_t, w^\star \rangle \in [-1, 1]$ tracks the alignment between the current direction $w_t \in \mathbb{S}^{d-1}$ and the planted direction $w^\star$, while $a_{k,t} := \langle f_t, h_k \rangle$ are the Hermite coefficients of the profile function $f_t \in L^2_\gamma$.

We distinguish two regimes depending on the index $s \in \mathbb{N}^\star$, which is defined as the smallest integer $s \geq 1$ such that $a_s^\star \neq 0$. In other words, $s$ corresponds to the leading nonzero mode in the Hermite expansion of the target function $\varphi^\star$. This parameter controls the initial signal strength in the direction dynamics, and determines both the presence and the duration of the transient phase before convergence. Finally, the coefficients that converge toward a zero target value exhibit exponentially fast convergence to zero, uniformly in time and independently of the dimension.

**Case $s \geq 2$. Flatness of the population risk around $m = 0$.** In this case, the first nonzero coefficient $a_s^\star$ appears at index $s \geq 2$. As a result, the initial gradient signal in the direction $w$ is weak, scaling like $m_0^{s-1}$. The theorem asserts the existence of a *concentration time* $\tau_c \geq m_0^{-2(s-1)} \sim d^{s-1}$, beyond which the system enters an exponential convergence phase. In fact, $\tau_c$ is the time needed for *weak recovery*, i.e. the time at which $m$ reaches a level set independent of the dimension. We actually prove in Lemma A.5 in the Appendix that there exist $b, B > 0$, such that $bd^{(s-1)} \leq \tau_c \leq Bd^{(s-1)}$, from which we deduce that $\tau_c \sim d^{s-1}$. After time $t \geq \tau_c$, the flow converges toward one of two symmetric attractors: (i) if $m_0 > 0$, then $m_t \to 1$, and $a_{k,t} \to a_k^\star$, or (ii) if $m_0 < 0$, then $m_t \to -1$, and $a_{k,t} \to (-1)^k a_k^\star$, with exponentially rates modulated by a sequence $b \in \ell_2(\mathbb{N})$, normalized as $\|b\|_{\ell_2} = 1$. The key phenomenon is that the joint dynamics enable recovery regardless of the sign of $m_0$. In contrast to the planted model, where negative initialization traps the flow in a spurious basin, here the adaptive evolution of the function $f_t$ corrects for early misalignment.

**Case $s = 1$. Immediate convergence.** In the special case $s = 1$, the first Hermite coefficient $a_1^\star$ is already active. This injects a strong signal into the direction dynamics from the outset, and no transient phase is needed. The convergence is exponential for all $t \geq 0$, with rate independent of dimension. The basin of attraction is now determined by the sign of the product $a_{1,0} a_1^\star$, i.e., whether the initial function $f_0$ shares the correct sign on the leading mode. If this product is positive, the system converges to $m_t \to 1$, and if negative, to $m_t \to -1$, with corresponding flips in the sign pattern of the Hermite coefficients.

The essential quantity that governs recovery is not the individual convergence of $m_t$ or $a_{k,t}$, but rather the convergence of their product: $a_{k,t} m_t^k \longrightarrow a_k^\star$, as $t \to \infty$. This is the quantity that enters into the effective representation $f_t(\langle w_t, x \rangle)$, and determines the predictive performance of the model. Since $m_t^k \to (\pm 1)^k$ and $a_{k,t} \to (\pm 1)^k a_k^\star$, we

obtain in both cases:

$$f_t(\langle w_t, x \rangle) = \sum_{k=0}^{\infty} a_{k,t} h_k(\langle w_t, x \rangle)$$

$$\longrightarrow \sum_{k=0}^{\infty} a_k^\star h_k(\langle w^\star, x \rangle) = \varphi^\star(\langle w^\star, x \rangle),$$

regardless of the sign of $m_0$. Thus, the flow achieves perfect recovery of the target regression function in the limit $t \to \infty$, even though the feature direction $w_t$ may converge to either $+w^\star$ or $-w^\star$, depending on the initialization.

### 4.3. Description of the dynamics and sketch of the proof

We now describe and break down the dynamics of the system distinguishing between the case $s \geq 2$ and the case $s = 1$, which behaves differently. Initially, recall that $m_t$ is very close to zero on a scale of order $1/\sqrt{d}$.

**Case $s \geq 2$.** The dynamics happens in different phases:

1. **Phase I: rapid movement of $a_{k,t}$ towards $a_k^\star m_t^k$.** On the one hand, in the dynamics of $m$, initially, the vector field is of intensity $m^{s-1}$ at first order. Hence, $m_t$ remains very close to the initialization for a time, at least of order $m_0^{1-s}$. On the other hand, given Eq. (8), the movement of $a_k$ corresponds to an exponentially fast shrinkage at rate 1 until it hits the neighborhood of $a_k^\star m_t^k$. Hence, during this period, an arbitrary number $k^*$ of coefficients $a_{k,t}$ partially stabilize toward $a_k^* m_t^k$ within a time $t_{k^*}$ of order $k^* \log(1/m_0)$. The slow dynamics of $m_t$, compared to the fast dynamics of $a_{k,t}$ (fast-slow dynamics), allows the system to achieve a *monotonicity principle*: Indeed, in light of equation Eq.(9), it is now clear to see that $m_t$ increases if $m_0$ is positive (respectively decreasing if $m_0$ is negative) from $t_{k^*}$. In fact, the terms corresponding to the indices larger than $k^*$ do not contribute to the dynamics, as they correspond to the remainder of a converging series.

2. **Phase II: slow increase of $m$.** The dynamics stabilizes in virtue of the *monotonicity principle* and for a constant $c$ independent of the dimension, we have $\dot{m}_t \sim cm^{2s-1}$ until the system reaches this constant. We deduce that reaching this constant takes a time of order $m_0^{-2(s-1)} \sim d^{s-1}$.

3. **Phase III: exponential convergence.** After that, the dynamics of the order parameter $m_t$ becomes independent of the initial value and converges exponentially fast as the $a_{k,t}$ follow the lead.

**Case $s = 1$.** This case works quite differently from the previous one. In virtue of Eq.(8), $a_{1,t}$ partially stabilizes towards $a_1^* m_t$. However, this time, $m_t$ follows the fast dynamics. Indeed, initially, $m_t$ will move linearly toward $a_0^* a_{1,0}$:

Thus, $m_t$ adapts to the sign of $a_1^* a_{1,0}$ without any time dependence on $m_0$, while $a_{1,t}$ will not be able to change its sign. After this, $m_t$ will no longer change its sign. After a time independent of the initial condition, the system reaches again a *monotonicity principle* as in the previous case.

## 5. Kernel Implementation via Hermite Functions

The procedure described above considers an idealized scenario, where the non-parametric regression is performed on the whole ambient space $L_\gamma^2$. We show that this can be practically carried by performing this non-parametric regression on a dense Reproducing Kernel Hilbert Space $\mathcal{H} \subset L_\gamma^2$, written as RKHS in short, see (Schaback & Wendland, 2006; Schölkopf & Smola, 2002) for an overview.

**A Hermite dense RKHS.** First, we construct such a RKHS space, adapted to the Hermite structure of our problem. To do this let us define $(\mathsf{c}_k)_{k \in \mathbb{N}} \in \ell_2(\mathbb{N})$ a positive sequence. Furthermore, we ask that $\sum_k k\mathsf{c}_k < +\infty$, that is to say that $\mathsf{c}$ has some fast decay at infinity, e.g. $\mathsf{c}_k = o(k^{-2})$. Then, we use the following construction.

**Proposition 5.1.** *For all sequences $(\mathsf{c}_k)_{k \in \mathbb{N}}$ defined as previously, the subspace*

$$\mathcal{H} := \big\{ f \in L_\gamma^2 \cap \mathcal{C}(\mathbb{R}^q) \, ; \|f\|_{\mathcal{H}}^2 := \sum \mathsf{c}_k^{-1} a_k^2 < +\infty \big\}$$

*defines a dense RKHS with kernel $K(x,y) = \sum_k \mathsf{c}_k h_k(x) h_k(y) = \langle \phi(x), \phi(y) \rangle_{\ell_2}$, where the equality stands in a point-wise sense and $\phi_k = \sqrt{\mathsf{c}_k} h_k$. Finally, we also define, for any $f, g \in \mathcal{H}$, the inner product $\langle f, g \rangle_{\mathcal{H}} := \sum_k \mathsf{c}_k^{-1} \langle f, h_k \rangle \langle g, h_k \rangle$.*

Note that similar RKHSs have already been considered for numerical computation of expectation with respect to Brownian paths, e.g. (Irrgeher & Leobacher, 2015) and statistical estimation of multi-index problems (Follain & Bach, 2024). We refer to the first cited paper for the construction as well as more on smoothness properties of such RKHSs. The RKHS norm $\|f\|_{\mathcal{H}}^2$ is thus of the form $\sum_k \mathsf{c}_k^{-1} \langle f, h_k \rangle^2$, and therefore defines a weighted Sobolev space, similar to $\mathcal{H}^p$, if $c$ decreases as a power law. Note also that it is possible to use a random process to express the kernel, as it has been already pinned as a *random feature expansion of the kernel* (Rahimi & Recht, 2007) (see (Bietti et al., 2023) for such a construction). Note that this could also lead to some regularization, by considering the loss $\mathcal{L}(f, w) + \mu \|f\|_{\mathcal{H}}^2/2$ that would act as a threshold of the high frequencies of the function $\varphi^\star$.

**Approximation with finite dimensional RKHS.** Obviously, one can define for any integer $\mathsf{k} \in \mathbb{N}^*$, a finite dimensional RKHS that we will denote $\mathcal{H}_{\mathsf{k}}$ by truncating the feature map $\phi$ as $\phi_{\mathsf{k}} = (\sqrt{c_0} h_0, \sqrt{c_1} h_1, \dots, \sqrt{c_{\mathsf{k}}} h_{\mathsf{k}}) \in \mathbb{R}^{\mathsf{k}+1}$. This

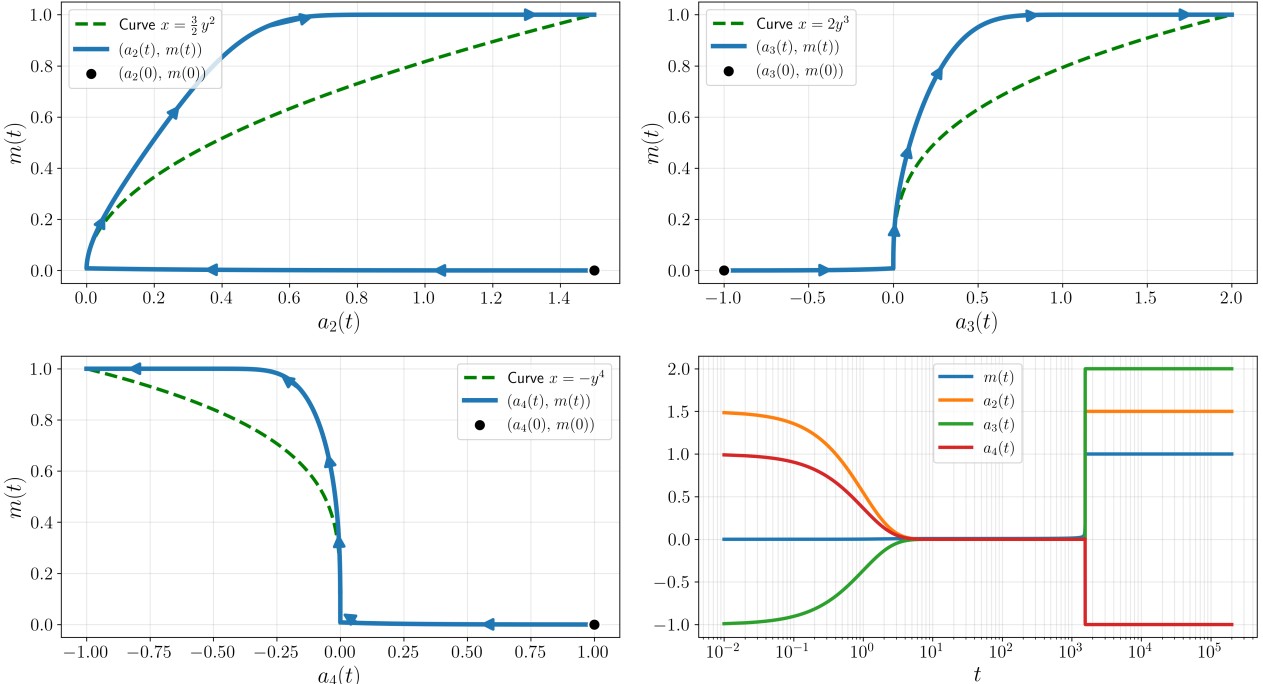

*Figure 1.* **Idealized dynamics** obtained by Euler discretization of the ODE system in Eqs. (8)–(9) for the planted function $\varphi^\star = \frac{3}{2} h_2 + 2 h_3 - h_4$ (so $s = 2$, $a_2^\star = 3/2$, $a_3^\star = 2$ and $a_4^\star = -1$). Initial conditions: $a_2(0) = 3/2$, $a_3(0) = -1$, $a_4(0) = 1$ and $m(0) = 10^{-4}$. **Top-left, top-right, bottom-left:** phase portraits in the $(a_k(t), m(t))$ planes for $k = 2, 3, 4$; arrows indicate the direction of time. The green dashed curves are the quasi-static equilibria $a_k(t) \simeq a_k^\star m(t)^k$ (i.e. $x = \frac{3}{2} y^2$, $x = 2 y^3$ and $x = -y^4$) predicted by the fast–slow analysis (Phase I, Section 4.3). Each trajectory tracks its algebraic curve as long as $m(t) \ll 1$, and departs from it at the onset of Phase III, when $m(t)$ crosses a critical threshold. **Bottom-right:** learning curves of $m(t)$ and of the coefficients $a_2(t)$, $a_3(t)$, $a_4(t)$ as functions of time (log scale on the time axis), showing the three regimes: fast decay of $a_k(t)$ towards $a_k^\star m(t)^k$ (Phase I), slow growth of $m(t)$ at quasi-static equilibrium (Phase II), and sharp convergence once $m(t)$ exceeds the critical threshold (Phase III).

enables in practice to run the gradient flow on $((a_k)_{k \leq \mathsf{k}}, m)$ as a true ODE, or a numerical method, by discretizing it via an Euler scheme that fits on a computer. Obviously, the larger the $\mathsf{k}$, the better the approximation will be as for any $f \in \mathcal{H}^1$,

$$
\begin{aligned}
\inf_{g \in \mathcal{H}_\mathsf{k}} \| f - g \|_{L_\gamma^2}^2 &= \sum_{k \geq \mathsf{k}+1} a_k^2 \\
&\leq \left( \sum_{k \geq \mathsf{k}+1} k^2 a_k^2 \sum_{k \geq \mathsf{k}+1} \frac{a_k^2}{k^2} \right)^{\frac{1}{2}} \\
&\leq \frac{\| f \|_{\mathcal{H}^1}^2}{\mathsf{k}^2}.
\end{aligned}
$$

This is what is done in the following section.

## 6. Numerical Experiments

To validate and illustrate our theoretical findings, we present two sets of numerical experiments. The first set directly discretizes the idealized gradient flow equations derived in the infinite-dimensional, population-risk setting, whereas the second set considers the empirical loss and finite-dimensional approximation described in Section 5. Both experiments aim to shed light on the characteristic fast–slow dynamics of the system and the role of the symmetry-breaking in the joint learning process.

**Simulation of the idealized dynamics.** In this first set of experiments, we directly integrate the idealized gradient-flow ODE on the scalar coefficients $a_{k,t}$ and the alignment parameter $m_t := \langle w_t, w^\star \rangle$ derived in the infinite-sample, population-risk setting (Eqs. (8)–(9)), via an explicit Euler scheme with step size $\Delta t = 10^{-2}$ over a time horizon $T = 2 \times 10^5$. The planted function is $\varphi^\star = \frac{3}{2} h_2 + 2 h_3 - h_4$, a linear combination of Hermite polynomials up to degree four (so the truncation level is $\mathsf{k} = 4$, the information exponent is $s = 2$, and the target coefficients are $a_2^\star = 3/2$, $a_3^\star = 2$, $a_4^\star = -1$). We initialize the correlation at a small positive value $m_0 = 10^{-4}$ — kept moderate so that the Phase II duration, which scales as $m_0^{-2(s-1)}$, remains tractable within the simulation horizon — to mimic the weak alignment typically induced by high-dimensional random

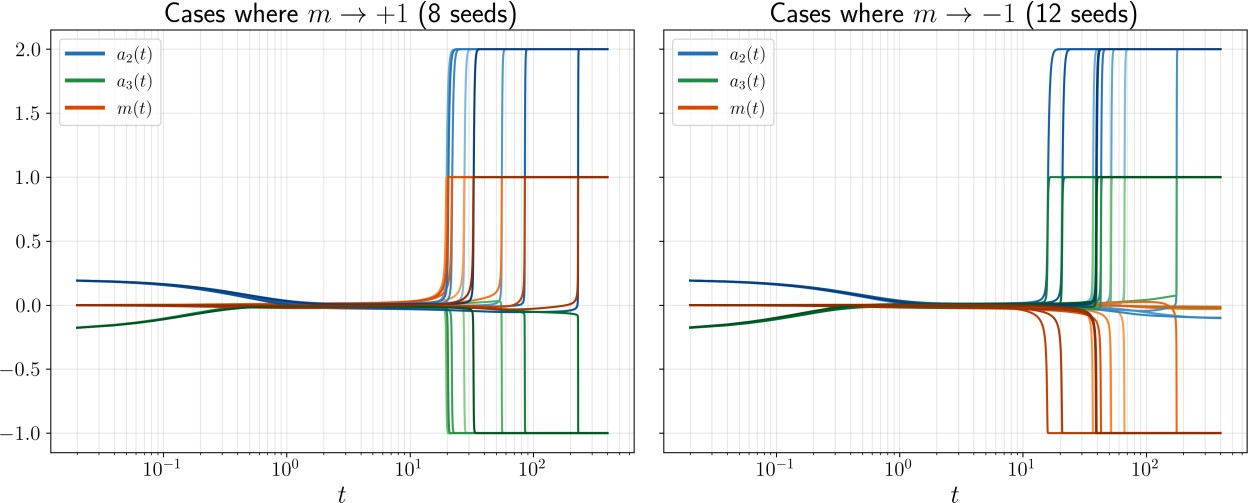

*Figure 2.* **Empirical RKHS dynamics.** Full-batch gradient descent on the empirical risk in the finite-dimensional RKHS $\mathcal{H}_k$ (with $k = 3$) described in Section 5, using $n = 10^5$ i.i.d. standard Gaussian inputs in dimension $d = 100$ and the planted function $\varphi^\star(x) = 2\, h_2(\langle w^\star, x\rangle) - h_3(\langle w^\star, x\rangle)$ (so $a_2^\star = 2$, $a_3^\star = -1$). Unlike Figure 1, which simulates the idealized ODE, this experiment runs the discrete-time, finite-sample algorithm actually implementable on a computer. Each panel displays the evolution of the coefficients $a_2(t)$, $a_3(t)$ and of the alignment $m(t) = \langle w(t), w^\star \rangle$ as a function of the continuous-time variable $t = \gamma \cdot \text{step}$ (log scale), so that the empirical and idealized dynamics share a common time axis. Because the planted function carries an odd Hermite component, the symmetry $w \mapsto -w$ is broken. We run 20 seeds and split them by the sign of the final alignment, displaying the case $m \to +1$ on the left and $m \to -1$ on the right. Every individual seed trajectory is plotted, with one color family per quantity ($a_2$ in blues, $a_3$ in greens, $m$ in oranges) and a distinct shade for each seed. We deliberately avoid mean $\pm$ standard-deviation bands: across seeds the trajectories are essentially time-translates of one another, differing only in the random activation time of Phase III, so any pointwise average would smear the sharp transition into a smooth ramp and hide the very phenomenon under study. The qualitative behavior matches the idealized dynamics of Figure 1: a first $O(1)$ decay of coefficients magnitude (Phase I), then a plateau (Phase II) followed by a sharp transition to recovery (Phase III).

initialization (e.g., when $w_0 \sim \text{Unif}(\mathcal{S}_{d-1})$ with $d \gg 1$). The coefficients are initialized to the deterministic values $a_2(0) = 3/2$, $a_3(0) = -1$, $a_4(0) = 1$, chosen with mixed signs so that the trajectory must cross the slow manifold from a generic position (no sign assumption is built into the initial state). Figure 1 displays, in a $2 \times 2$ layout, the phase portraits of the dynamics in the $(a_k(t), m(t))$ planes for $k = 2, 3, 4$, together with the full learning curves of $m(t)$ and of all three coefficients $a_2(t), a_3(t), a_4(t)$ as functions of time (with a logarithmic scale on the time axis to make all three phases visible simultaneously). We observe a sharp transition in behavior, in line with the theory developed for the case $s \geq 2$. Note that as long as $m, a_k \ll 1$, each phase portrait perfectly follows its quasi-static curve $a_k(t) \simeq a_k^\star m(t)^k$ (in green in Figure 1), showing that, over this stage of duration of order $d^{s-1}$, the dynamics reach a partial equilibrium in the $a$-variables.

**Empirical RKHS-based dynamics.** In the second set of experiments, we implement the joint learning procedure in the practical setting described in Section 5. The function class is represented by a finite-dimensional truncation $\mathcal{H}_k$ of the Hermite-based RKHS with $k = 3$, and the empirical risk is minimized via standard *full-batch* gradient

descent. We generate a dataset of $n = 10^5$ i.i.d. Gaussian inputs in dimension $d = 10^2$, with labels computed from the planted function $\varphi^\star(x) = 2\, h_2(\langle w^\star, x\rangle) - h_3(\langle w^\star, x\rangle)$ (so $a_2^\star = 2$, $a_3^\star = -1$, and we choose $w^\star = e_1$), and use a constant step size $\gamma = 2 \times 10^{-2}$ over $T = 2 \times 10^4$ iterations (effective continuous time $\gamma T = 400$, well past the predicted transition). Both the direction $w_t$ and the coefficients $a_{k,t}$ are updated jointly; $w_t$ is re-projected on the unit sphere after each step. The direction is initialized by drawing $v \sim \mathcal{N}(0, I_d)$. Note that we decide to replace its first coordinate by $\pm\epsilon$ with a uniformly random sign ($\epsilon = 10^{-5}$), and normalizing $w_0 = v/\|v\|$. This allows to control de order of magnitude of the alignement at initialization to prevent having to deal with too long runs. The coefficients are initialized at $a_{2,0} = 0.2$, $a_{3,0} = -0.2$. Because the planted function contains an odd Hermite component, the symmetry $w \mapsto -w$ is broken and the random sign of the warm start selects which of the two fixed points $m = \pm 1$ is reached. We therefore run 20 independent seeds and split them into the two groups according to the sign of the final alignment. Figure 2 displays, on a logarithmic time axis and for each of the two cases, the evolution of the alignment $m(t)$ and of the coefficients $a_2(t)$, $a_3(t)$. Rather than re-

porting summary statistics, we plot every individual seed trajectory, using a separate color family per quantity ($a_2$ in blues, $a_3$ in greens, $m$ in oranges) with a distinct shade for each seed. This choice is deliberate: while the trajectories agree closely in *shape*, the random duration of the Phase II plateau induces a seed-dependent activation time at which Phase III ignites, so the curves look like *time-translates* of one another. Consequently, the pointwise $L^2$ or $L^\infty$ distance between two trajectories can be large during the transition window even though their qualitative behavior is identical, and aggregating with mean $\pm$ standard deviation would smear the sharp transition into a gentle ramp, hiding precisely the fast–slow phenomenon under study. This observation also has a methodological consequence for the analysis: a uniform-in-time bound between the empirical trajectory and the idealized ODE would necessarily be conservative, since it must absorb this random activation-time shift; a direct, phase-by-phase argument — as we develop in the theoretical part — would be far tighter. With this caveat in mind, the qualitative behavior closely matches the idealized dynamics described earlier, suggesting that with enough samples and a small enough step size, the empirical dynamics tracks the idealized one up to a random time-shift. These experiments confirm that the key phenomena predicted in the population-risk regime persist in the finite-sample setting, even under stochastic initialization and discretization. They also validate the practical relevance of the RKHS-based modeling approach for analyzing learning in single-index neural architectures.

## 7. Conclusion and Outlook

In this work, we have analyzed the dynamics of a joint gradient flow designed to learn a univariate function of a linear projection. The evolution of the system exhibits a striking separation of timescales between the nonparametric learning of the scalar function and the estimation of the projection direction. This separation gives rise to a fast–slow dynamical regime, especially pronounced in high dimensions and for complex target functions. A particularly intriguing outcome of our analysis is that joint learning, despite its apparent lack of structural knowledge, can outperform scenarios in which the target function is fixed *a priori*. In some cases, fixing the function can trap the dynamics in a suboptimal basin, whereas the flexibility of joint learning allows the system to escape such traps and achieve global convergence. This behavior echoes a broader phenomenon observed in overparameterized models, such as neural networks, where learning flexibility often aids in finding better minima.

With the temporal structure of the flow now well understood, a natural next step is to investigate the *sample complexity* of joint learning (Damian et al., 2024; Maillard et al., 2020). Our analysis has so far focused on the infinite-sample (pop-

ulation) regime, but practical implementations rely on finite datasets (Mei et al., 2018), with a finite truncation of the Hermite expansion and a discrete-time algorithm. In such settings, both the estimation of the functional coefficients $a_{k,t}$ and the evolution of the correlation parameter $m_t$ are impacted by statistical noise and generalization error. Quantifying the number of samples required to reliably recover the target coefficients and ensure convergence is thus an essential direction for future work. In particular, it is important to understand how this sample complexity depends on the initialization $m_0$, the number of learned coefficients, and the spectral structure of the target (Ben Arous et al., 2022). Such an analysis would bridge the gap between the idealized continuous-time dynamics studied here and the finite-data, discrete-time algorithms employed in practice.

**Acknowledgments.** A. S. acknowledges support from the Deutsche Forschungsgemeinschaft (DFG, German Research Foundation) through project number 432176920, as well as support from the ERC MSCA grant SLOHD (101203974). L. P.-V. acknowledges support from the Fondation Bézout, the support of Hi! PARIS and ANR/France 2030 program (ANR-23-IACL-0005) through the grant GALA. Finally, the authors extend their gratitude to the *Incubateur de Fraîcheur* for providing an ideal atmosphere that fostered exceptional discussions.

## Impact Statement

This work is purely theoretical and has no foreseeable direct societal impact.

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

# A. Appendix

The appendix is organized as follows:

- We first present the proof of Lemma 2.1 in Section A.1, which concerns the form of the loss.

- In Section A.2, we prove Lemma 3.1, which describes the dynamics of the Hermite coefficients and the order parameter $m$.

- At the beginning of Section A.3, we look at the limitation of the dynamics for the planted model and prove Proposition 4.1.

- The rest of Section A.3 is dedicated to the proof of the joint dynamics, which corresponds to Theorem 4.2.

In the following, the constants $c$ and $C$ denote dimension-independent constants whose values may change from line to line throughout the proofs.

## A.1. Problem Setup

We begin by proving Lemma 2.1 about the expansion of the loss.

*Proof of Lemma 2.1.* We first recall that

$$\mathcal{L}(f, w) = \frac{1}{2}\mathbb{E}\left[(f(\langle w, X\rangle) - \varphi^\star(\langle w^\star, X\rangle) - \varepsilon)^2\right]$$
$$= \frac{1}{2}\|f\|_{L^2_\gamma}^2 + \frac{1}{2}\|\varphi^\star\|_{L^2_\gamma}^2 + \frac{1}{2}\mathrm{Var}(\varepsilon) - \mathbb{E}\left[(f(\langle w, X\rangle)\varphi^\star(\langle w^\star, X\rangle)\right],$$

by developing the square and using the fact that $\langle w, X\rangle \sim \langle w^\star, X\rangle \sim \mathcal{N}(0, 1)$ and that $\varepsilon$ is independent of $X$. We recall that our analysis is based on the *Hermite polynomials* $(h_k)_{k\geq 0}$, which form a complete orthonormal basis and we deduce that

$$\|f\|_{L^2_\gamma}^2 = \sum_{k=0}^{+\infty} a_k^2 .$$

In addition, one readily check that

$$\mathrm{Cov}(\langle w, X\rangle, \langle w^\star, X\rangle) = \langle w, w^\star\rangle =: m ,$$

and by a simple covariance calculation, we obtain

$$\mathbb{E}\left[f\left(\langle w, X\rangle\right)\varphi^\star\left(\langle w^*, X\rangle\right)\right] = \mathbb{E}_{Y,Z}\left[f\left(Y\right)\varphi^\star\left(mY + \sqrt{1-m^2}Z\right)\right]$$
$$= \mathbb{E}_Y\left[f\left(Y\right)\mathbb{E}_Z\varphi^\star\left(mY + \sqrt{1-m^2}Z\right)\right] ,$$

where $Y, Z$ are i.i.d. standard Gaussian. A fundamental fact is that the operator

$$T_m[g](y) = \mathbb{E}_Z\left[g\left(my + \sqrt{1-m^2}Z\right)\right]$$

is diagonal in the *Hermite polynomials* basis and we have

$$T_m[h_i] = m^i h_i .$$

Thus, by writing $f, \varphi^\star$ in the the Hermite polynomials basis, we obtain

$$\mathbb{E}\left[f\left(\langle w, X\rangle\right)\varphi^\star\left(\langle w^*, X\rangle\right)\right] = \mathbb{E}_Y\left[f\left(Y\right)\sum_{i=0}^{\infty} a_i^* T_m[h_i](Y)\right]$$
$$= \sum_{i=0}^{\infty}\sum_{k=0}^{\infty} a_i^* a_k m^i \mathbb{E}_Y\left[h_k(Y)h_i(Y)\right]$$
$$= \sum_{k=0}^{\infty} a_k a_k^* m^k ,$$

the proof follows. □

In this paper, we are interested in two different gradient flows. Their description is the subject of the next section.

### A.2. Gradient Flow Dynamics

**Gradient flow on $L_\gamma^2 \times \mathcal{S}_{d-1}$.** In this section, we derive the equations of motions of the gradient flow on $\mathcal{L}(f, w)$. We say that $(f_t, w_t)_{t \geq 0}$ follow the gradient flow if they are solution of the infinite dimensional system of ODEs

$$\frac{\mathrm{d}}{\mathrm{d}t} f_t = -\nabla_f^{L_\gamma^2} \mathcal{L}(f_t, w_t) ,$$

$$\frac{\mathrm{d}}{\mathrm{d}t} w_t = -\nabla_w^{\mathcal{S}_{d-1}} \mathcal{L}(f_t, w_t) .$$

This corresponds to the infinite system of ODEs stated in the Lemma 3.1 and proven below.

*Proof of Lemma 3.1.* We recall the spherical gradient $\nabla_w^{\mathcal{S}_{d-1}} u(w) = \nabla_w u(w) - \langle w, \nabla_w u(w) \rangle w$ for any differentiable function $u$. Thus, one readily checks that Eq. (6) can be written as

$$\frac{\mathrm{d}}{\mathrm{d}t} w_t = -(Id - w_t w_t^T) \nabla_w \mathcal{L}(f_t, w_t) .$$

Using Eq. (4), we obtain

$$\frac{dm_t}{dt} = \frac{\mathrm{d}(w^\star)^T w_t}{\mathrm{d}t} = (w^\star)^T (Id - w_t w_t^T) \sum_{i=1}^{\infty} a_{i,t} a_i^* i m_t^{i-1} w^\star = \left(1 - m_t^2\right) \sum_{i=1}^{\infty} i a_{i,t} a_i^* m_t^{i-1} .$$

Then, we use the Hilbert property of $(L_\gamma^2, \langle \cdot, \cdot \rangle_{\mathcal{L}_\gamma^2})$ and Riesz theorem to the represent the Fréchet differential of $\mathcal{L}(\cdot)$ in terms of a gradient, i.e. for all $g \in L_\gamma^2$,

$$\left\langle \nabla_f^{L_\gamma^2} \mathcal{L}(f, w), g \right\rangle_{L_\gamma^2} = \lim_{h \to 0} \frac{\mathcal{L}(f + hg, w) - \mathcal{L}(f, w)}{h} .$$

A simple calculation leads to

$$\mathcal{L}(f + hg, w) = \mathcal{L}(f, w) + \frac{h^2}{2} \mathbb{E}\left(g^2(\langle X, w \rangle)\right) + h\mathbb{E}((f(\langle X, w \rangle) - \varphi^*(\langle X, w^\star \rangle))g(\langle X, w \rangle)) ,$$

and we have

$$\mathbb{E}((f(\langle X, w \rangle) - \varphi^*(\langle X, w^\star \rangle))g(\langle X, w \rangle))) = \mathbb{E}_Y((f(Y) - T_m[\varphi*](Y))g(Y))$$

$$= \mathbb{E}_Y((f(Y) - \sum_{k=0}^{+\infty} a_k^* m^k h_k(Y))g(Y))$$

$$= \langle f - \sum_{k=0}^{+\infty} a_k^* m^k h_k, g \rangle ,$$

from which we deduce that

$$\nabla_f^{L_\gamma^2} \mathcal{L}(f, w) = f - \sum_{k=0}^{+\infty} a_k^* m^k h_k = \sum_{k=0}^{+\infty} \left(a_k - a_k^* m^k\right) h_k .$$

We recall that *Hermite polynomials* $(h_k)_{k \geq 0}$ form a complete orthonormal basis and we deduce that

$$\frac{da_{k,t}}{dt} = a_k^* m_t^k - a_{k,t}, \; \forall k \geq 0, \, t \geq 0$$

with Eq. (5) . □

We conclude this section with the proof of Lemma 3.2.

*Proof of Lemma 3.2.* This is a consequence of the following Lemma

**Lemma A.1.** *For all $a \geq 0$ and $0 < \varepsilon < 1$, we have*

$$2\Big(1 - F\big(a\sqrt{1+\varepsilon}\big)\Big) \ - \ 2e^{-d\varepsilon^2/4} \ \leq \ \mathbb{P}\big(\sqrt{d}|\langle w_0, w^\star\rangle| > a\big) \ \leq \ 2\Big(1 - F\big(a\sqrt{1-\varepsilon}\big)\Big) \ + \ 2e^{-d\varepsilon^2/4}, \tag{11}$$

*where $F$ is the distribution function of the one dimensional standard Gaussian.*

*Proof.* By rotation invariance of the law of $w_0$, we can represent $w_0 = G/\|G\|$, where $G \sim \mathcal{N}(0, I_d)$ and hence $\sqrt{d}\langle w_0, w^\star\rangle$ has the same law as $\frac{g_1}{\sqrt{\frac{1}{d}\sum_{i=1}^d g_i^2}}$. Let us call $S = \frac{1}{d}\sum_{i=1}^d g_i^2$, then

$$\begin{aligned}
\mathbb{P}(|g_1|/\sqrt{S} \geq a) &= \mathbb{P}(|g_1|/\sqrt{S} \geq a \,|\, S \in [1-\varepsilon, 1+\varepsilon])\, \mathbb{P}(S \in [1-\varepsilon, 1+\varepsilon]) \\
&\qquad\qquad + \mathbb{P}(|g_1|/\sqrt{S} \geq a \,|\, S \notin [1-\varepsilon, 1+\varepsilon])\, \mathbb{P}(S \notin [1-\varepsilon, 1+\varepsilon]) \\
&\leq \mathbb{P}(|g_1| \geq a\sqrt{1-\varepsilon}) + \mathbb{P}(|S - 1| \geq \varepsilon)\,.
\end{aligned}$$

But now, the first term is exactly equal to $2\Big(1 - F\big(a\sqrt{1-\varepsilon}\big)\Big)$, and for the second, we use concentration of subexponential variables. Then, we follow the classical proof of concentration presented in (Vershynin, 2018), that is, for all $t \geq 0$ and all $\lambda \in (0, 1/2)$,

$$\begin{aligned}
\mathbb{P}(S - 1 \geq \varepsilon) = \mathbb{P}\left(\sum_{i=1}^d (g_i^2 - 1) \geq \varepsilon d\right) &\leq e^{-\lambda \varepsilon d} \prod_{i=1}^d \mathbb{E}\exp(\lambda(g_i^2 - 1)) \\
&= e^{-\lambda d(\varepsilon + 1)}(1 - 2\lambda)^{-d/2} \\
&= e^{-\lambda d(\varepsilon + 1)} e^{-d/2 \ln(1 - 2\lambda)} \\
&\leq e^{-\lambda d(\varepsilon + 1)} e^{\frac{d\lambda}{1 - 2\lambda}}\,.
\end{aligned}$$

We optimize and choose $\lambda = \frac{1}{2}(1 - \frac{1}{\sqrt{1+\varepsilon}})$, that yields:

$$\mathbb{P}(S - 1 \geq \varepsilon) \leq e^{-\frac{d}{2}(1 - \sqrt{\varepsilon + 1})^2} \leq e^{-\frac{d\varepsilon^2}{4}}\,.$$

A totally symmetric argument yields the inequality on the other direction. $\qquad\square$

Now we directly bound:

$$\begin{aligned}
\mathbb{P}(a \leq \sqrt{d}|\langle w_0, w^\star\rangle| \leq b) &= 1 - \mathbb{P}(\{\sqrt{d}|\langle w_0, w^\star\rangle| > b\} \cup \{\sqrt{d}|\langle w_0, w^\star\rangle| < a\}) \\
&\geq 1 - \mathbb{P}(\sqrt{d}|\langle w_0, w^\star\rangle| > b) - \mathbb{P}(\sqrt{d}|\langle w_0, w^\star\rangle| < a) \\
&= \mathbb{P}(\sqrt{d}|\langle w_0, w^\star\rangle| > a) - \mathbb{P}(\sqrt{d}|\langle w_0, w^\star\rangle| > b) \\
&\geq 2\left(F\left(\frac{b}{\sqrt{2}}\right) - F\left(\frac{\sqrt{3}}{\sqrt{2}}a\right)\right) - 4e^{-d/16} \\
&\geq 2\left(1 - \frac{e^{-b^2/4}}{b\sqrt{\pi}} - \frac{1}{2} - \frac{a\sqrt{3}}{2\sqrt{\pi}}\right) - 4e^{-d/16} \\
&\geq 1 - a - \frac{2e^{-b^2/4}}{b\sqrt{\pi}} - 4e^{-d/16}\,,
\end{aligned}$$

where we used $F(x) \geq 1 - \frac{e^{-x^2/2}}{x\sqrt{2\pi}}$ and the Taylor expansion of $F$ at order 1 for the second estimate. $\qquad\square$

We now turn to the convergence of the flow and the advantage of a coupled dynamics.

## A.3. Convergence Behavior of the Joint Gradient Flow

**The Planted Model.** In this case, the dynamics reduce to a one-dimensional ordinary differential equation for the correlation $m_t := \langle w_t, w^\star \rangle$, as previously derived

$$\dot{m}_t = (1 - m_t^2) \sum_{k=1}^{\infty} k(a_k^\star)^2 m_t^{k-1}.$$

We show here the Proposition 4.1 about the planted model. We restate it here for the sake of clarity:

**Proposition 1.**

- If $m_0 > 0$, the trajectory converges to alignment: $\lim_{t \to \infty} m_t = 1$.

- If $m_0 < 0$, then, (i) if $s \geq 3$ is odd, we have $\lim_{t \to \infty} m_t = 0$;

  else, (ii) if $s$ is even, then

  - If $\bar{m} > 1$, the trajectory converges to anti-alignment: $\lim_{t \to \infty} m_t = -1$.
  - If $\bar{m} < 1$, the trajectory remains trapped in a local minimizer and: $\lim_{t \to \infty} m_t = -\bar{m}$.

*Proof of Proposition 4.1.* If $m_0 > 0$, it is clear that $\dot{m}_t(0) > 0$. By continuity, $m_t$ strictly increases until $(1 - x^2)\Phi(x) = 0$ for $x > 0$, which happens only for $x = 1$, where the system then stabilizes: This follows by combining $\dot{m}_t \geq (1 - m_t^2)s(a^*)^2(m_0)^{s-1}$ with Gronwall's Lemma and $m_t \leq 1$.

If $m_0 < 0$, and $s \geq 3$ is odd, we have $\dot{m}_t(0) > 0$. By continuity, $m_t$ increases until $(1 - x^2)\Phi(x) = 0$ for $x > m_0$, which corresponds to $x = 0$ for $m_0$ arbitrary small. By Picard–Lindelöf theorem, we have uniqueness of the solutions and thus $m_t < 0$. We deduce that $\dot{m}_t \geq c(m_t)^{s-1}$, which lead to the convergence to 0 by Gronwall's Lemma.

If $s$ is even, then , we have $\dot{m}_t(0) < 0$. By continuity, $m_t$ decreases until $(1 - x^2)\Phi(x) = 0$ for $x \leq m_0$, which corresponds to $x = -\bar{m}$ if $\bar{m} < 1$ or $x = -1$ if $\bar{m} > 1$. In the case where $\bar{m} < 1$, by Picard–Lindelöf theorem, $m_t > -\bar{m}$ and we conclude by noting that $m_t$ cannot converge to a larger $\alpha > -\bar{m}$. If it did, as $m_t$ is decreasing, $\alpha$ should be a zero of $\Phi(x)$. The case $\bar{m} > 1$ follows by a similar argument.

$\square$

Proposition 4.1 highlights the limitations of the planted model. The joint dynamics does not suffer from such issues.

### A.3.1. MAIN THEOREM: CONVERGENCE IN THE JOINT LEARNING SETTING

We now move on to solving the system of differential equations (8)-(9). Note that if $a_k^* = 0$, $a_{k,t}$ tends exponentially fast to 0 by (8) and the dynamics of $m_t$ is independent of such $a_{k,t}$. Therefore, from now on, we are interested only in the $a_{k,t}$ such that $a_k^* \neq 0$. *In passing*[1], we mention that $f_t \in L_\gamma^2$ for all $t \geq 0$. Indeed, the following states that the $a_{k,t}$ remain within $[-\max(|a_k^*|, |a_{k,0}|), |\max(|a_k^*|, |a_{k,0}|)]$. Indeed,

*Fact* 1. We have $|a_{k,t}| \leq \max(|a_k^*|, |a_{k,0}|)$. If $|a_{k,t}| \leq |a_k^*|$ for some $t$, then for $t' \geq t$, $|a_{k,t'}| \leq |a_k^*|$. Finally, If it does not exist $t$ such that $|a_{k,t}| \leq |a_k^*|$, then $a_{k,t}$ tends exponentially fast to $a_k^*$ (monotonically). Hence, for all $t \geq 0$, $f_t \in L_\gamma^2$ and the flow is well defined at all times.

*Proof.* Using (8), it follows that

$$\frac{da_{k,t}}{dt} \geq 0 \iff a_k^* m_t^k \geq a_{k,t} \implies \begin{array}{l} a_k^* \geq a_{k,t} \text{ if } a_k^* > 0. \\ -a_k^* \geq a_{k,t} \text{ if } a_k^* < 0, \end{array} \implies |a_{k*}| \geq a_{k,t}$$

by using the fact that $|m_t| \leq 1$ and in the same way, $\frac{da_{k,t}}{dt} \leq 0$ implies that $a_{k,t} \geq -|a_k^*|$. We conclude the proof of the first two points. We now address the third point. For $m_0 > 0$ and $a_{k,0} > a_k^* > 0$, we have

$$\frac{da_{k,t}}{dt} \leq a_k^* - a_{k,t},$$

---

[1]as we say in the London Society Club of French Gentlemen.

and by Grönwall's lemma, we obtain that

$$a_{k,t} \le a_k^* + (-a_k^* + a_{k,0})e^{-t}.$$

By hypothesis $a_{k,t} \ge a_k^*$, which concludes the proof for this case. The other cases work in the same way. □

We are now in a position to establish the main theorem.

*Proof of Theorem 4.2.* We divide the proof of the theorem into two subsections, the case $s \ge 2$ and the case $s = 1$. The remainder of the appendix is devoted to the proof of this Theorem: The argument proceeds through several key lemmas, which we state and prove below before completing the proof of the theorem.

A.3.2. PROOF OF THEOREM 4.2 FOR $s \ge 2$

In all the following, we define for all $\kappa \in (0,1)$, the time at which $m$ attains level set $\kappa > 0$.

$$\tau_\kappa = \inf\{t \ge 0 \,|\, |m_t| \ge \kappa\}.$$

Denote $u_{k,t} = a_{k,t} - a_k^* m_t^k$ and write $\bar{m}_{[t,t_0]} = \sup_{t_0 \le s \le t} |m_s|$. The goal of the next lemmas is to show that, after a time of order $\log(1/m_0)$, $m_t$ has barely moved, while an arbitrary number of $a_{k,t}$ are nearly equal to $a_k^* m_t^k$.

**Lemma A.2.** *There exists $C > 0$, such that for all $k \ge s$, and all $t \ge t_0$, we have*

$$|a_{k,t} - a_k^* m_t^k| \le C\bar{m}_{[t,t_0]}^{k+s-2} + |a_{k,t_0} - a_k^* m_{t_0}^k|e^{-(t-t_0)} \tag{12}$$

*Proof.* We have for all $k \ge s, t \ge 0$,

$$\dot{u}_{k,t} = \dot{a}_{k,t} - ka_k^* m_t^{k-1}\dot{m}_t$$
$$= -u_{k,t} - ka_k^* m_t^{k-1}(1 - m_t^2)\sum_{k \ge s} ka_{k,t}a_k^* m_t^{k-1}.$$

That is to say that

$$|\dot{u}_{k,t} + u_{k,t}| = \left| ka_k^* m_t^{k-1}(1-m_t^2)\sum_{j \ge s} ja_{j,t}a_j^* m_t^{j-1} \right|$$
$$\le k|a_k^*||m_t|^{k-1}(1-m_t^2)\sum_{j \ge s} j|a_{j,t}a_j^*||m_t|^{j-1}$$
$$\le k|a_k^*||m_t|^{k-1}|m_t|^{s-1}\sum_{j \ge s} j|a_{j,t}a_j^*||m_t|^{j-s},$$

and using $|a_{j,t}| \le |a_{j,0}| + |a_j^*|$, we get

$$|\dot{u}_{k,t} + u_{k,t}| \le k|a_k^*||m_t|^{k+s-2}\sum_{j \ge s} j(|a_j^*|^2 + |a_{j,0}a_j^*|)$$
$$\le k|a_k^*||m_t|^{k+s-2}\sum_{j \ge s} j(2|a_j^*|^2 + |a_{j,0}|^2)$$
$$\le C|m_t|^{k+s-2},$$

where $C$ is a uniform in $k$ upperbound on $k|a_k^*|\sum_{j \ge s} j(|a_j^*|^2 + |a_{j,0}a_j^*|)$. Applying the Gronwall inequality between $t$ and $t_0$, we have that

$$|u_{k,t} - u_{k,t_0}e^{-(t-t_0)}| \le Ce^{-t}\int_{t_0}^t |m_u|^{k+s-2}e^u du \le C\bar{m}_{[t,t_0]}^{k+s-2},$$

which ends the proof of the lemma. □

Let $k^* \geq s$ an integer that we consider large enough, with no dependency on the dimension though. Define $t_{k^*} = Ck^* \ln(1/m_0)$. The first step is to establish that $m_t$ stays near its initial value $m_0$ for a duration of order $m_0^{1-s} >> t_{k^*}$ and that $\dot{m}_{t_{k^*}}$ has the sign of the initial value.

**Lemma A.3.** *At time $t_{k^*}$, for all $k \leq k^*$, we have $a_{k,t_{k^*}} a_k^* m_{t_{k^*}}^k > 0$. Moreover,*

- *if $m_0 > 0$, then $Cm_0 > m_t > cm_0$ for $t \leq \frac{c}{m_0^{s-1}}$ and $\dot{m}_{t_{k^*}} > 0$,*

- *if $m_0 < 0$, then $-Cm_0 < m_t < -cm_0$ for $t \leq \frac{c}{m_0^{s-1}}$ and $\dot{m}_{t_{k^*}} < 0$.*

Then, after time $t_{k^*}$, the $a_{k,t}$ for $k \leq k^*$ have concentrated near their equilibrium $a_k^* m_t^k$ and, therefore it is possible to reinforce the lemma A.2 with the following version.

**Lemma A.4.** *There exist constants $C, c > 0$, such that for all $s \leq k \leq k^*$, and all $t \geq t_k^*$, we have*

$$|a_{k,t} - a_k^* m_t^k| \leq C\bar{m}_{[t,t_{k^*}]}^{k+2s-2}. \tag{13}$$

*Proof of Lemma A.3.* The first step is to establish that $m_t$ stays near its initial value $m_0$ for a duration of order $m_0^{1-s} >> t_{k^*}$. We first consider $m_0, a_s^* > 0$ and define $\tau = \inf\{t \geq 0, m_t \geq Cm_0 \text{ or } m_t \leq cm_0\}$. For $t \leq \tau$, by using (8), we obtain

$$ca_s^* m_0^s - a_{2,t} \leq \frac{da_{s,t}}{dt} \leq Ca_s^* m_0^s - a_{2,t},$$

from which we deduce that

$$ca_s^* m_0^s + (-a_s^* cm_0^s + a_{s,0})e^{-t} \leq a_{s,t} \leq a_s^* Cm_0^s + (-a_s^* Cm_0^s + a_{s,0})e^{-t}.$$

The previous inequality can be readily adapted to various cases: In the case where $a_s^* m_0^s < 0$, we interchange $C$ and $c$. In our case, by using (9) for $t \leq \tau$, we have

$$ce^{-t}m_0^{s-1} + cm_0^s - C\sum_{k=s+1} m_t^{k-1} \leq \frac{dm_t}{dt} \leq Ce^{-t}m_0^{s-1} + Cm_0^s + C\sum_{k=s+1} m_t^{k-1}.$$

All in all, we get

$$cm_0^{s-1} - Cm_0^s t + m_0 \leq m_t \leq Cm_0^{s-1} + Cm_0^s t + m_0,$$

from which we conclude that

$$\frac{c}{m_0^{s-1}} \leq \tau \leq \frac{C}{m_0^{s-1}}.$$

Meanwhile, we now prove that $a_{k,t}$ is the sign of $a_k^* m_0^k$ at $t_{k^*}$. Let $a_k^* > 0$, by using (8), we get that for $t \leq \tau$

$$\frac{da_{k,t}}{dt} \geq a_k^* (cm_0)^k - a_{k,t}, \tag{14}$$

and by Gronwall's lemma, we have

$$a_{k,t} \geq a_k^* (cm_0)^k + (-a_k^* (cm_0)^k + a_{k,0})e^{-t},$$

which proves that $a_{k,t}$ is strictly positive exponentially fast. More precisely, $a_{k,t} \geq a_k^* (cm_0)^k/2$ in a time smaller than

$$\log\left(\frac{2(a_k^* (cm_0)^k - a_{k,0})}{a_k^* (cm_0)^k}\right),$$

which is order $k \log(1/m_0) << cm_0^{1-s}$. The case $a_k^* < 0$ works the same way by noting that the inequality (14) is true in the opposite direction. The case where $m_0$ is negative is very similar and is left to the reader. Thus, at time $t_{k*}$, $m_{t_{k*}} > 0$ and $a_{k,t_{k*}} a_k^* m_{t_{k*}}^k > 0$. In addition, we have

$$\frac{dm_t}{dt} = \left(1 - m_t^2\right) \sum_{i=s}^{k^*} i a_{i,t} a_i^* m_t^{i-1} + \left(1 - m_t^2\right) \sum_{i=k^*+1}^{+\infty} i a_{i,t} a_i^* m_t^{i-1} > 0 \text{ for } t = t_k^*,$$

since all the terms in the first sum are positive and the first summand is at least of order $m_0^{2s-1}$ while the second sum is at most of order $m_0^{k*}$. $\square$

*Proof of Lemma A.4.* From equality Eq.(12), taken between $0$ and $t \geq t_{k*}$, we have

$$\begin{aligned} |a_{k,t}| &\leq |a_k^* m_t^k| + C\bar{m}_{[t,0]}^{k+s-2} + Cm_0^{Ck^*} \\ &\leq |a_k^* \bar{m}_{[t,0]}^k| + C\bar{m}_{[t,0]}^{k+s-2} \\ &\leq C|\bar{m}_{[t,0]}^k| \\ &\leq C|\bar{m}_{[t,t_{k*}]}^k| \,. \end{aligned}$$

The last inequality by lemma (A.3). Then, we estimate

$$\begin{aligned} \sum_{j \geq s} j |a_{j,t} a_j^*| |m_t|^{j-1} &\leq C|m_t|^{s-1} \bar{m}_{[t,t_{k*}]}^s \sum_{j \geq s} j \bar{m}_{[t,t_{k*}]}^{j-s} |a_j^*| |m_t|^{j-s} \\ &\leq C\bar{m}_{[t,t_{k*}]}^{2s-1} \sum_{j \geq s} j |a_j^*| |\bar{m}_{[t,t_{k*}]}|^{2(j-s)} \\ &\leq C\bar{m}_{[t,t_{k*}]}^{2s-1} \,. \end{aligned}$$

Notice that in this estimation, we have hence gained a factor $s$ compared to before. This gives

$$\begin{aligned} |\dot{u}_{k,t} + u_{k,t}| &= \left| k a_k^* m_t^{k-1}(1 - m_t^2) \sum_{j \geq s} j a_{j,t} a_j^* m_t^{j-1} \right| \\ &\leq k|a_k^*| |m_t|^{k-1} \sum_{j \geq s} j |a_{j,t} a_j^*| |m_t|^{j-1} \\ &\leq C|m_t|^{k-1} \bar{m}_{[t,t_{k*}]}^{2s-1} \\ &\leq C\bar{m}_{[t,t_{k*}]}^{k+2s-2} \,, \end{aligned}$$

that we integrate between $t_{k*}$ and $t$. Then, the end of the proof follows the same way as before. $\square$

Hence we can define

$$\begin{aligned} \tilde{\tau}_{k*} &= \inf\{t \geq t_{k*} \,|\, \forall k \leq k^*, \, a_{k,t_{k*}} a_k^* m_{t_{k*}}^k \leq 0 \text{ or } \dot{m}_t \text{ changes sign}\} \\ &= \inf\{t \geq t_{k*} \,|\, \forall k \leq k^*, \, a_{k,t} \text{ changes sign or } a_k^* m_t^k \text{ changes sign or } \dot{m}_t \text{ changes sign}\} \,. \end{aligned} \tag{15}$$

We will prove eventually that $\tilde{\tau}_{k*} = +\infty$. Before this, we prove quantitatively that there exists a constant $c > 0$, so that after a time of order $m_0^{2(1-s)}$, $m$ reaches level $c$.

**Lemma A.5.** *There exists $b, B > 0$, such that $b/m_0^{2(s-1)} \leq \tau_c \leq B/m_0^{2(s-1)}$, moreover $\tilde{\tau}_{k*} \geq \tau_c$.*

We postpone the proof of Lemma A.5 for after. By the inequality Eq.(12), as $m$ has reached a constant level $c > 0$, this means that all $k \leq k^*$, this is also the case for the $a_{k,(\cdot)}$. Let us define the constants, for all $k \leq k^*$,

$$c_k = \frac{1}{2} \min\{|a_{k,\tau_c}|, |a_k^* m_{\tau_c}^k|\} \,, \text{ and the times } \bar{\tau}_k = \inf\{t \geq \tau_c \,|\, |a_{k,t}| \leq c_k\}.$$

**Lemma A.6.** *For all $k \leq k^*$, we have $\bar{\tau}_k = \tilde{\tau}_{k^*} = +\infty$.*

*Proof.* To fix the ideas, we fix a $k \leq k^*$ and assume $a_k^* > 0$ as well as the case $m_0 > 0$. For all $t \in [\tau_c, \min_k \bar{\tau}_k]$, $\dot{m}_t > 0$ if $\sum_{k \geq s} k a_{k,t} a_k^* m_t^{k-1} > 0$, but we instantly see, that this is the case splitting the sum before $k^*$ which as a positive and constant contribution and after that can be as small as we want by choosing a large $k^*$. Hence, $m$ increases strictly in $[\tau_c, \min_k \bar{\tau}_k]$. Furthermore, $\dot{a}_{k,t} = a_k^* m_t^k - a_{k,t}$, so that either $a_{k,t} \geq a_k^* m_t^k > a_k^* m_{\tau_c}^k > c_k$ or in the other case $\dot{a}_{k,t} > 0$, and hence $a_{k,t} \geq \min\{a_k^* m_{\tau_c}^k, a_{k,\tau_c}\} > c_k$. Either case, if $\min_k \bar{\tau}_k < \infty$, this leads to a contradiction. Finally, as we have seen that $m$ increases strictly in $[\tau_c, \min_k \bar{\tau}_k]$ and that all $a_{k,(\cdot)}$ are lower bounded by a constant, $\tilde{\tau}_{k^*} \geq \min_k \bar{\tau}_k = +\infty$. $\square$

We move to the convergence of the coefficients $(a_{k,t})$ and $m_t$ for $s \geq 2$, thereby addressing the first part of Theorem 4.2.

**Theorem A.7.** *There exists constants $c, C > 0$ so that after time $t \geq \tau_c \geq \dfrac{C}{m_0^{2(s-1)}}$,*

- *if $m_0 > 0$,*
$$|1 - m_t| \leq Ce^{-c(t-\tau_c)} ,$$
$$|a_{k,t} - a_k^*| \leq C b_k e^{-c(t-\tau_c)} , \text{ for all } k \in \mathbb{N} ,$$

- *if $m_0 < 0$,*
$$|1 + m_t| \leq Ce^{-c(t-\tau_c)} ,$$
$$|a_{k,t} - (-1)^k a_k^*| \leq C b_k e^{-c(t-\tau_c)} , \text{ for all } k \in \mathbb{N} ,$$

*where, $b \geq 0$ is a normalized sequence of $\ell_2(\mathbb{N})$, i.e. $\|b\|_{\ell_2(\mathbb{N})} = 1$.*

*Proof.* Let us first show the contraction of $m$. Assume that we are in the case where $m_0 > 0$, define $v_t = 1 - m_t \geq 0$, for all $t \geq \tau_c$, we have

$$\dot{v}_t = -(1 - m_t^2) \sum_{k \geq s} k a_{k,t} a_k^* m_t^{k-1} \leq -v_t \left(2c - \sum_{k \geq k^*} k |a_{k,t} a_k^* m_t^{k-1}|\right) ,$$

but once again this latter sum can be made arbitrarily small choosing a large $k^*$, so that

$$\dot{v}_t \leq -cv_t ,$$

and the first inequality comes from Gronwall lemma. For the contraction of the $a$, we pose $z_{k,t} = |a_{k,t} - a_k^*|^2$, and for all $t \geq \tau_c$,

$$\frac{d}{dt} z_{k,t} = 2(a_{k,t} - a_k^*)(a_k^* m_t^k - a_{k,t})$$
$$= -2z_{k,t} + 2a_k^*(a_{k,t} - a_k^*)(m_t^k - 1)$$
$$= -2z_{k,t} + 2a_k^*(a_{k,t} - a_k^*)(m_t - 1) \sum_{i=1}^{k} m^i ,$$

so that we have

$$|\dot{z}_{k,t} + 2z_{k,t}| \leq Ck|a_k^*|\sqrt{z_{k,t}} e^{-c(t-\tau_c)}$$

Defining $\bar{z}_{k,t} = \sqrt{z_{k,t+\tau_c}}$, we have that is satisfies, for all $t \geq 0$,

$$|\dot{\bar{z}}_{k,t} + \bar{z}_{k,t}| \leq C b_k e^{-ct},$$

where $\|b\|_{\ell_2(\mathbb{N})} = 1$ which integrates in close form and gives a solution that is upper bounded by $\bar{z}_{k,t} \leq C b_k e^{-ct}$, and this finishes the proof of the theorem in the case $m_0 > 0$.

For the sake of clarity we draw the lines of the case $m_0 < 0$ but there are essentially the same. Define $v_t = 1 + m_t \geq 0$, for all $t \geq \tau_c$, we have

$$\dot{v}_t = v_k(1 - m_t) \sum_{k \geq s} k a_{k,t} a_k^* m_t^{k-1} \leq v_t \left(-2c + \sum_{k \geq k^*} k |a_{k,t} a_k^* m_t^{k-1}|\right) ,$$

but once again this latter sum can be made arbitrarily small choosing a large $k^*$, so that

$$\dot{v}_t \leq -cv_t \,,$$

and the first inequality comes from Gronwall lemma. For the contraction of the $a$, if $k$ is even, we pose $z_{k,t} = |a_{k,t} - a_k^*|^2$, and for all $t \geq \tau_c$,

$$\frac{d}{dt} z_{k,t} = -2z_{k,t} + 2a_k^*(a_{k,t} - a_k^*)(m_t + 1)(m_t - 1) \sum_{i=1}^{k/2} m^{2i} \,,$$

so that we have

$$|\dot{z}_{k,t} + 2z_{k,t}| \leq Ck|a_k^*|\sqrt{z_{k,t}} e^{-c(t-\tau_c)}$$

and the proof ends as before. Now, if $k$ is odd, we pose $z_{k,t} = |a_{k,t} + a_k^*|^2$, and for all $t \geq \tau_c$,

$$\frac{d}{dt} z_{k,t} = -2z_{k,t} + 2a_k^*(a_{k,t} + a_k^*)(m_t^k + 1)$$

$$= -2z_{k,t} + 2a_k^*(a_{k,t} + a_k^*)(m_t + 1) \sum_{i=1}^{k} (-1)^i m^i$$

so that we have

$$|\dot{z}_{k,t} + 2z_{k,t}| \leq Ck|a_k^*|\sqrt{z_{k,t}} e^{-c(t-\tau_c)}$$

and the proof ends, once again, as before. This final case concludes the proof of the theorem. $\qquad\square$

We finish this section by proving Lemma A.5.

*Proof of Lemma A.5.* Let us assume first that $m_0 > 0$, the case where $m_0 < 0$, is symmetric by multiplying by $-1$ (and thus reverting) all inequalities below. Then, after time $t_{k^*}$, by Lemma A.3, we have $m_{t_{k^*}} > 0$ as well as $\dot{m}_{t_{k^*}} > 0$. By continuity of all processes involved, we know that $\tilde{\tau}_{k^*} > t_{k^*}$. Then, for all $t \in [t_{k^*}, \tilde{\tau}_{k^*}]$, as $m$ increases, we have $m_t = \bar{m}_{[t,t_{k^*}]}$ and

$$\dot{m}_t = (1 - m_t^2) \sum_{k \geq s} k a_{k,t} a_k^* m_t^{k-1}$$

$$\geq \frac{3}{4} \sum_{k=s}^{k^*} k a_{k,t} a_k^* m_t^{k-1} + \frac{3}{4} \sum_{k \geq k^*+1} k a_{k,t} a_k^* m_t^{k-1}$$

$$\geq \frac{3}{4} s a_{s,t} a_s^* m_t^{s-1} - \frac{3}{4} \sum_{k \geq k^*+1} k |a_{k,t} a_k^* m_t^{k-1}| \,.$$

From there, by Lemma A.4 applied between $t_{k^*}$ and $t$, we have

$$a_s^* m_t^s - C\bar{m}_{[t,t_{k^*}]}^{3s-2} \leq a_{s,t} \leq a_s^* m_t^s + C\bar{m}_{[t,t_{k^*}]}^{3s-2} \,,$$

and hence we have

$$a_{s,t} a_s^* m_t^{s-1} \geq |a_s^*|^2 m_t^{2s-1} - C|a_s^*| m_t^{3s-1} \,,$$

so that

$$\dot{m}_t \geq \frac{3s|a_s^*|^2}{4} m_t^{2s-1} - \frac{3sC|a_s^*|}{4} m_t^{3s-1} - \frac{3}{4} \sum_{k \geq k^*+1} k |a_{k,t} a_k^* m_t^{k-1}|$$

$$\geq b m_t^{2s-1} - bC m_t^{3s-1} - Cm_t^{k^*} \sum_{k \geq k^*+1} k |a_{k,0} a_k^*|$$

$$\geq b m_t^{2s-1} - bC m_t^{3s-1}$$

$$\geq b m_t^{2s-1}(1 - C m_t^{s-1}) \,.$$

Hence, for $s \geq 2$, there exists a constant $c > 0$, such that for all $t \in [t_{k^*}, \tilde{\tau}_{k^*} \wedge \tau_c]$,

$$\dot{m}_t \geq cm_t^{2s-1} \, .$$

Let us assume that $\tilde{\tau}_{k^*} < \tau_c$, then either

- $m_{\tilde{\tau}_{k^*}} = 0$, which is impossible because in virtue of the inequality above $m$ is increasing between $t_{k^*}$ and $\tilde{\tau}_{k^*}$.

- $\dot{m}_{\tilde{\tau}_{k^*}} = 0$, and hence $m_{\tilde{\tau}_{k^*}} = 0$ which is impossible because of the bullet point above.

- $a_{k,\tilde{\tau}_{k^*}} = 0$, which is impossible because for all $t \in [t_{k^*}, \tilde{\tau}_{k^*} \wedge \tau_c]$, in the case $a_k^* > 0$,

$$\dot{a}_{k,\tilde{\tau}_{k^*}} = a_k^* m_{\tilde{\tau}_{k^*}}^k > a_k^* m_0^k > 0 \, ,$$

which is a contraction since $a_{k,t}$ cannot cross from positive to negative with a positive speed. The case where $a_k^* < 0$ is exactly similar.

Hence, finally, $\tilde{\tau}_{k^*} \geq \tau_c$, and for all $t \in [t_{k^*}, \tau_c]$, $\dot{m}_t \geq cm_t^{2s-1}$. This means that $\tau_c \leq t_{k^*} + C/m_0^{2(s-1)}$.

For the lower bound, this rests on the same argument as for $t \geq t_{k^*}$,

$$\begin{aligned}
\dot{m}_t &= (1 - m_t^2) \sum_{k \geq s} k a_{k,t} a_k^* m_t^{k-1} \\
&\leq s a_{s,t} a_s^* m_t^{s-1} + m_t^s \sum_{k \geq s+1} k a_{k,t} a_k^* m_t^{k-s} \\
&\leq b m_t^{2s-1} + b C m_t^{2s} \\
&\leq C m_t^{2s-1} \, ,
\end{aligned}$$

up to a constant level $c > 0$, if $c$ is a small enough constant. This ends the proof of the lemma. $\qquad\square$

### A.3.3. PROOF OF THEOREM 4.2 FOR $s = 1$

The case $s = 1$ is somewhat different. In the case $s \geq 2$, $m_t$ initially remains very close to $m_0$, allowing the coefficients $a_{k,t}$ to adopt the sign of their target values (the sign of $a_k^* m_t^k$). In contrast, in the case $s = 1$, $m_t$ initially moves linearly toward the sign of $a_{1,0} a_1^*$ in an initial phase and only after that do the coefficients $a_{k,t}$ adopt the sign of their target values: the sign of $a_k^* m_t^k$.

**Lemma A.8.** *There exists a time $T \geq C$ independent of $m_0$, such that both $m_T$ and $a_{1,T}$ are bounded away from zero by a constant independent of $m_0$. Moreover, $m_T$ has the sign of $a_{1,0} a_1^*$, while $a_{1,T}$ has the sign of $a_{1,0}$.*

*Proof.* We prove that $a_{1,t}$ does not have time to reach 0, as $m_t$ is already far from 0 in the direction of $a_1^* a_{1,0}$. Let

$$T_1 = \inf\{t \geq 0, |a_{1,t} - a_{1,0}| \geq \frac{|a_{1,0}|}{2}\}.$$

By using (8), for all $t \leq T_1$, we obtain that

$$|a_{1,t} - a_{1,0}| \leq \int_0^t \left| \frac{da_{1,u}}{du} \right| du \leq \int_0^t |a_1^*| + |a_{1,u}| \, du \leq t(|a_1^*| + \frac{3}{2}|a_{1,0}|).$$

We deduce $T_1 \geq |a_{1,0}|/(2(|a_1^*| + \frac{3}{2}|a_{1,0}|))$, which is of order 1. Meanwhile, for $t \leq T_1 \wedge \tau_{1/2}$, we have that

$$c_1 a_1^* a_{1,0} - C|m_t| \leq \frac{dm_t}{dt} \leq c_2 a_1^* a_{1,0} + C|m_t|$$

where $c_1 = 1/2, c_2 = 3/2$ if $a_1^* a_{1,0} > 0$ and the contrary else. Let

$$T_2 = \inf\{t \geq 0, |m_t| > \min(1/2, |a_1^* a_{1,0}|/4C)\}.$$

We thus deduce that for $t \leq T_1 \wedge T_2 =: T$,

$$c_3 a_1^* a_{1,0} t + m_0 \leq m_t \leq c_4 a_1^* a_{1,0} t + m_0,$$

where $c_3, c_4 > 0$. We deduce that after a time of order $m_0$, $m_t$ is a sign of $a_{1,0} a_1^*$. At time $T$, $m_t$ and $a_{1,t}$ are of order 1 and both have the sign of $a_{1,0}$. $\qquad \square$

We now prove that, after time $T$, both $m_t$ and $a_{1,t}$ remain uniformly bounded away from zero.

**Lemma A.9.** *There exists constants $c, C > 0$ so that after time $t \geq C$, $|m_t|, |a_{1,t}| \geq c$. Moreover,*

- *if $a_1^* a_{1,0} > 0$, then $m_t > c$,*

- *if $a_1^* a_{1,0} < 0$, then $m_t < -c$.*

*Proof.* Without loss of generality, we consider $a_{1,0} < 0$, and $a_1^*, m_0 > 0$. From $t \geq T$, we recall that $a_{1,t}, m_t < 0$ and of order 1. However, from here, $m_t$ is free to increase or decrease for a certain time. We now prove that $m_t$ is however bounded away from 0 by a constant $-\delta$ independent of $m_0$. By defining

$$\tilde{T}_\delta = \inf\{t \geq T, a_1^* m_t \geq -\delta\},$$

for $-\delta > m_T$ small enough: First, we require $\frac{a_{1,0}}{2} < -\delta$ to have $a_{1,T} < -\delta$. Then, we claim that $a_1^* m_t$ must reach $-\delta$ before $a_{1,t}$ does. Indeed, by recalling that

$$\frac{da_{1,t}}{dt} = a_1^* m_t - a_{1,t}, t \geq 0,$$

we deduce that in order to hit $-\delta$, $a_t$ has to be in an increasing phase, that is $a_1^* m_t \geq a_{1,t}$ from which we deduce that $a_1^* m_t$ must reach $-\delta$ before $a_{1,t}$. We first prove that $\tilde{T}_\delta \geq -c \log(\delta)$. Indeed for $\tilde{T}_{\sqrt{\delta}} \leq t \leq \tilde{T}_\delta$ (we require also $\frac{a_{1,0}}{2} < -\sqrt{\delta}$ ), we have

$$\frac{da_{1,t}}{dt} \leq -\delta - a_{1,t},$$

which implies that for $t \geq \tilde{T}_{\sqrt{\delta}}$

$$a_{1,t} \leq -\delta + (a_{1,\tilde{T}_{\sqrt{\delta}}} + \delta)e^{-(t - \tilde{T}_{\sqrt{\delta}})} \leq -\delta + (-\sqrt{\delta} + \delta)e^{-(t - \tilde{T}_{\sqrt{\delta}})},$$

from which we deduce that $\tilde{T}_\delta \geq -c \log(\delta)$. We recall that for $t \leq \tau_{1/2}$, it exists $C$ such that

$$\frac{dm_t}{dt} \leq a_1^* a_{1,t} + 2a_2^* a_{2,t} m_t + C m_t^2,$$

and thus we obtain

$$\frac{dm_t}{dt}\Big|_{t=\tilde{T}_\delta} \leq a_1^* a_{1,\tilde{T}_\delta} - 2a_2^* a_{2,\tilde{T}_\delta} \delta/a_1^* + C(\delta/a_1^*)^2 \leq -a_1^* \delta - 2a_2^* a_{2,\tilde{T}_\delta} \delta/a_1^* + C(\delta/a_1^*)^2.$$

To conclude, we consider an adversarial case where $a_2^* > 0$ and $a_{2,t} < 0$, and we deduce that $\frac{dm_t}{dt}\big|_{t=\tilde{T}_\delta}$ is negative because $a_{2,t} \geq -(a_1^*)^2/(4a_2^*)$ in a time independent of $\delta$ (see $\frac{da_{2,t}}{dt} \leq -a_{2,t}$). $\qquad \square$

We now prove that an arbitrary number of coefficients $a_{k,t}$ adopt the sign of their target values (the sign of $a_k^* m_t^k$).

**Lemma A.10.** *For $k^* > 0$, it exists $t_{k^*} \geq C$, for all $k \leq k^*$, we have $a_{k,t_{k^*}} a_k^* m_{t_{k^*}}^k > 0$.*

*Proof.* The proof is very similar to the proof of Lemma A.3. We now return to our prototypical example : Let $a_{1,0} < 0$, and $a_1^*, m_0 > 0$. Consider $a_{2k}^* > 0$, using Lemma A.9, for $t \geq T$, we have

$$\frac{da_{2k,t}}{dt} \geq a_{2k}^* \delta^{2k} - a_{2k,t},$$

which implies, by Gronwall's lemma, that

$$a_{2k,t} \geq a_{2k}^* \delta^{2k} + (-a_{2k}^* \delta^{2k} + a_{2k,0})e^{-t},$$

which proves that $a_{2k,t}$ is strictly positive exponentially fast. The other cases work similarly. Thus, at time $t_{k^*}$, $m_{t_{k^*}} < 0$ by Lemma A.9 and $a_{k,t_{k^*}} a_k^* m_{t_{k^*}}^k > 0$ for $k \leq k^*$. $\qquad\square$

We move to the convergence of $m_t, (a_{k,t})$.

**Theorem A.11.** *There exists constants $c, C > 0$ so that after time $t \geq 0$,*

- *if $a_1^* a_{1,0} > 0$,*
$$|1 - m_t| \leq Ce^{-ct},$$
$$|a_{k,t} - a_k^*| \leq Cb_k e^{-ct}, \text{ for all } k \in \mathbb{N},$$

- *if $a_1^* a_{1,0} < 0$,*
$$|1 + m_t| \leq Ce^{-ct},$$
$$|a_{k,t} - (-1)^k a_k^*| \leq Cb_k e^{-ct}, \text{ for all } k \in \mathbb{N},$$

*where, $b \geq 0$ is a normalized sequence of $\ell_2(\mathbb{N})$, i.e. $\|b\|_{\ell_2(\mathbb{N})} = 1$.*

*Proof.* Assume that we are in the case where $a_{1,0} < 0$, and $a_1^*, m_0 > 0$. We recall that

$$\frac{dm_t}{dt} = \left(1 - m_t^2\right) \sum_{i=s}^{k^*} i a_{i,t} a_i^* m_t^{i-1} + \left(1 - m_t^2\right) \sum_{i=k^*+1}^{+\infty} i a_{i,t} a_i^* m_t^{i-1}.$$

At time $t = t_k^*$, by Lemma A.9 and Lemma A.10, the first sum on the right-hand side of the latter equation is negative. Using (8), we immediately see that $a_{k,t}$, $k \leq k^*$ cannot change sign for $t \geq t_k^*$: The derivative $\frac{da_{k,t}}{dt}$ has the sign of $a_k^* m_t^k$ for $a_{k,t} = 0$. Define $v_t = 1 + m_t \geq 0$, for all $t \geq t_{k^*}$, we have

$$\dot{v}_t = (1 - m_t^2) \sum_{k \geq s} k a_{k,t} a_k^* m_t^{k-1} \leq v_t \left(-c + \sum_{k \geq k^*} k |a_{k,t} a_k^* m_t^{k-1}|\right),$$

because $a_{1,t} < -c$. But once again this latter sum can be made arbitrarily small choosing a large $k^*$, so that

$$\dot{v}_t \leq -cv_t,$$

and the first inequality comes from Gronwall lemma. For the contraction of $a$'s, we do exactly as in the proof of Theorem A.7. $\qquad\square$

The proof of Theorem 4.2 follows. $\qquad\square$

