# OpenReview forum: "Joint Learning in the Gaussian Single Index Model"
_ICML.cc/2026/Conference — ICML 2026 spotlight_

### Official Review · Reviewer_wCiD · 2026-03-06

**Soundness:** 4
**Presentation:** 3
**Significance:** 3
**Originality:** 4
**Overall Recommendation:** 5
**Confidence:** 4

**Summary:**

The authors analyse a learning algorithm for Gaussian single-index models of the form $\varphi^\star(\langle w^\star, x\rangle)$. It consists in the joint optimization of the projection direction $w \in \mathbb{S}^{d-1}$ via spherical gradient flow, together with the optimization of the link function $\varphi$ via gradient flow on the infinite-dimensional space of square-integrable functions. The analysis provides a description of the regimes of gradient flow, identifying three phases in terms of the behaviour of the order parameters, and then formulating predictions on learning times that depend on the information exponent of $\varphi^\star$. A comparison with the already well-studied planted model ($\varphi = \varphi^\star$) reveals that the joint learning setting is more robust to initialization, always leading to convergence to the global optimum, whereas the planted model is susceptible to getting stuck in suboptimal stationary points of the landscape with positive probability.

Finally, the authors provide an approximate kernel implementation of the method and run simulations that are compatible with the theoretical findings.

**Compliance With Llm Reviewing Policy:**

Affirmed.

**Final Justification:**

The rebuttal addressed all my questions and I believe that the proposed fix are suitable, hence I stand by my positive evaluation.

**Key Questions For Authors:**

- In Section 7, the authors mention that the main next direction is to establish sample complexity estimates. I agree that this is a very important question and I would like to ask the authors to comment on the technical challenges they expect to encounter. Thinking for example of online SGD as in *Ben Arous et al. (2021)* as the most straightforward route to studying sample complexity: is the difficulty primarily a matter of tracking many coupled equations, or could there be fundamental obstacles that do not appear in the planted case? In particular, could the fast–slow structure of the dynamics interact non-trivially with finite-sample noise during the slow phase?

- At a conceptual level, there is an analogy between gradient flow on $\varphi$ in $L^2$ and gradient-based optimization of wide or deep networks that approximate $\varphi^\star$. How far do the authors think this analogy can hold? If one substitutes $\varphi$ with $\tilde{\varphi}$ given by the output of a large neural network and performs gradient flow on it (in a spirit similar to the kernel approach of Sections 5 and 6), would the three-phase dynamical structure and the robustness to negative initialization be expected to persist, or are these features specific to the $L^2$ setting?

**Limitations:**

yes

**Strengths And Weaknesses:**

## Strengths

- The setting considered, with the link function directly optimized via gradient flow in $L^2$ jointly with the dynamics of the projection direction, is quite novel in this framework and provides a fresh and original perspective on single-index models.

- The phases of gradient flow dynamics described in Section 4.3 deepen the understanding of single-index models. Classical works such as *Ben Arous et al. (2021)* delineated only two phases (search and ballistic). In this model, the different learning algorithm *splits* the first phase in two: first the rapid movement of $a_{k,t}$, then the slow increase of $m$. Clear characterizations of this type, that rigorously connect properties of the architecture and learning algorithm to well-defined dynamical phases, are valuable in providing intuition that can then be tested on more complex models.

- The comparison with the planted model provides a clear example of how an "expert" algorithm with prior information on the task (the exact knowledge of $\varphi^\star$) can be outperformed by a more flexible model with many more parameters to be fitted. This is a commonly observed phenomenon in deep learning practice, but theoretical models exhibiting this property cleanly are not so easy to find.

- This study shows that joint learning is powerful and flexible in solving single-index models without requiring specific assumptions on $\varphi^\star$, even when both layers are trained simultaneously via gradient-based methods rather than separately as in related literature. This opens up the application of joint learning techniques to other theoretical models.

- The kernel implementation via the Hermite-based RKHS provides a practical counterpart to the abstract method analysed theoretically. The fact that the empirical phenomenology agrees with the theoretical predictions strengthens the claims of the theory section and indicates that they are robust to finite-dimensional approximation.

- The exposition is clear and accessible. The simplicity of the model allows the reader to absorb the findings without being overwhelmed by technicalities.



## Weaknesses

- Only population gradient flow is considered. As pointed out by the authors in Section 7, this does not allow one to study the sample complexity of the model. This limitation is particularly relevant since, given the originality of the learning rule, it is not straightforward to predict how it will behave under an empirically approximated risk function. The experimental section does not fully address this point either, since the number of samples used in Figures 1 and 2 is very large and no error bars are shown.

- It is not clear to me what the left plot in Figure 1 adds to the analysis. Could the authors clarify? The blue trajectory should include arrows to indicate the direction of time flow. Regarding the curve $x = -y^4$, the paper claims that "as long as $m, a_4 \ll 1$, the phase portrait perfectly follows" that line, but the plot displays the full $[-1,1] \times [0,1]$ rectangle with rather thick lines, making it difficult to distinguish the behaviour near the origin. I would suggest either modifying the figure with thinner lines (perhaps showing several pairs $(m, a_i)$ together to appreciate the different slopes) or considering removing it, since the right plot already illustrates the dynamics effectively.

### Minor points

- Line 215 (right column): it would be helpful to clarify that there could be another global minimum at $m = -1$ in the common case of even $\varphi^\star$.

- The paragraph starting at line 271 (left column) is redundant. The information exponent is reintroduced without explicit connection to its first definition one page earlier.

- Line 228 (right column): I find the naming "Delayed convergence due to weak initialization" somewhat misleading for the case $s \ge 2$. While the gradient signal is indeed weaker at initialization, this phrasing suggests that a stronger initialization could resolve the issue, which with overwhelming probability cannot happen. It may be more natural to connect $s \ge 2$ to the flatness of the population risk around $m = 0$, as is commonly done in the single-index literature.

- Figure 2 appears before Figure 1 ?!

---

> ### Author Rebuttal · Authors · 2026-03-26
>
> We thank Reviewer wCiD for their thorough and insightful review. We address each point below.
>
> **Weakness 1: Population regime and sample complexity.**
>
> We provide a detailed explanation of this in the answer of reviewer HV7h (see Weakness 2 / Question 4:) and refer to it. We stress here rapidly that the reason of the difficulty lies in a rigorous treatment of the interaction between SGD noise and the fast-slow structure — during the slow phase, the signal-to-noise ratio in is very low, and finite-sample noise could interfere with the monotonicity principle. Our heuristic is that this interaction is in fact *benign*, for the following reason: the fast-slow decoupling naturally tempers the stochastic fluctuations at each phase.
>
> Figure 2 already provides empirical evidence that the predicted dynamical structure persists under finite samples $( n = 10^5, d = 100)$; we will add error bars and further experiments in the revision.
>
> **Weakness 2: Figure 1 (left panel).**
>
> We agree with the reviewer's suggestions. The left panel displays the phase portrait of $(m_t, a_{4,t})$: the curve $a_{4,t} \approx a_4^\star m_t^4$ is the quasi-static equilibrium of Phase I, and the trajectory confirms this prediction as long as $m_t \ll 1$. We will improve the figure by: (i) adding arrows for the direction of time, (ii) using thinner lines and a zoom near the origin to make the tracking of the algebraic curve visible, and (iii) overlaying several pairs $(m_t, a_{i,t})$ to illustrate the different slopes $a_k^\star m_t^k$ for different $k$.
>
> **Minor points.**
>
> - *Line 215 ($m = -1$ as global minimum for even $\varphi^\star$).* Indeed, when $\varphi^\star$ is even, $m = -1$ is also a global minimizer (with $a_{k,t} \to (-1)^k a_k^\star = a_k^\star$). We will add this remark.
> - *Line 271 (redundant reintroduction of the information exponent).* We will remove the redundancy and refer back to the first definition.
> - *Line 228 ("Delayed convergence due to weak initialization").* We agree that this phrasing is misleading — it suggests the issue could be resolved by a better initialization, which is not the case with overwhelming probability. We will replace it with language connecting $s \geq 2$ to the *flatness of the population risk around $m = 0$*, as is standard in the literature.
> - *Figure ordering.* We will fix the ordering so that Figure 1 appears before Figure 2.
>
> **Key Question 1: Sample complexity**
>
> This question is addressed in *Weakness 1* above and detailed in *Weakness 2 / Question 4: Population regime and finite-sample extension* of **Reviewer HV7h**.
>
> **Key Question 2: Analogy with neural networks.**
>
> This is a natural and important question. To be precise: we do not analyze neural networks, and any direct claim would be far-fetched. The structural parallel is that our model has two layers — the projection $w$ and the function $f$ in the Hermite basis — but we optimize $f$ in $L^2_\gamma$ (or an RKHS), not via a neural parameterization.
>
> That said, we can comment on what we expect. If one replaces the $L^2_\gamma$ optimization with gradient flow on a neural network $\varphi_\theta$, the Hermite coefficients $a_{k,t} = \langle \varphi_{\theta_t}, h_k \rangle$ still satisfy equations of the same *form* as (8), but with additional nonlinear coupling through the parameterization. In the mean-field / lazy-training regime (wide networks), the dynamics of the Hermite coefficients decouple approximately, and one would expect the three-phase structure and robustness to negative initialization to persist. In the feature-learning regime (finite width), the situation is less clear: the network may learn a different basis than Hermite, and the decoupling that underpins our analysis would break down. Whether the qualitative conclusion (joint learning escapes traps) still holds in this regime is a compelling open question. Note that there has been attempts in this direction but the main problem of this studies is that the algorithm is totally changed (and handcrafted) for some analysis to hold  [Berthier et al. 2023, Bietti et al. 2023, Abbe et al. 2021]. The best comparison is the article on the XOR problem [Glasgow, 2023], but the analysis is very specific to the XOR and does not generalize beyond.

---

> > ### Author Rebuttal · Reviewer_wCiD · 2026-04-01
> >
> > The rebuttal addressed all my questions and I believe that the proposed fix are suitable, hence I stand by my positive evaluation.

---

### Official Review · Reviewer_tJw7 · 2026-03-08

**Soundness:** 4
**Presentation:** 4
**Significance:** 3
**Originality:** 3
**Overall Recommendation:** 5
**Confidence:** 3

**Summary:**

This articles studies __joint learning__ in the __Gaussian single-index model__, where the target is $x\mapsto \phi(\langle w_* ,x\rangle) $, and both the direction $w\in \mathbb{S}^{d-1}$, and the fonction $f\in L^{2}(\mathbb{R};\gamma)$ are trained by population gradient flow.

Using Hermite expansions, the loss can be summarized in terms of Hermite coefficients $(a_k)$ of the trained function $f$, and the coefficients $(a_k^{*})$ of the teacher $\phi$ and the overlap between the signal and the learned $w$. This __reduces the infinite-dimensional flow to a coupled ODE__ on $(a_{k,t},m_t)$.

 The main result is that joint learning __avoids the failure modes of the planted dynamic__ under negative initialization i.e. $m_0<0$. One other interesting aspect is that this dynamics is reminiscent of slow-fast dynamic, in the sense that the coefficients $a_{k,t}$ follow a fast dynamics given by
$$\frac{d}{dt}a_{k,t}= a_{k} m_{t}^{k}-a_{k,t}$$
 which goes to $$a_{k}^{*}m_{t}^{k},$$
and then other variables follow a slow evolution.

Finally, the authors propose a practical implementation via a Hermite-adapted RKHS truncation and demonstrate simulations consistent with the predicted phases.

**Compliance With Llm Reviewing Policy:**

Affirmed.

**Final Justification:**

I believe my concerns have been adequately addressed by the rebuttal. I keep my score.

**Key Questions For Authors:**

* The paper repeatedly invokes a “variational principle,” but this notion is never clearly defined. It would be helpful to state explicitly what is meant by this principle and how it is used in the analysis.

* Can you add a brief justification for the Gradient flow in $L^{2}_{\gamma}$ to be defined? (It might be naive, but a simple Cauchy-Lipschitz argument should be sufficient)

* The proof might benefit from being rephrased in the language of two-timescale expansions. Could the authors comment on whether their argument can be recast in such a framework?

* What is the relevance of this result for sample-based considerations, since, even if the SGD can unfortunately fail, I still can simulate $n$ initializations and keep the best of those $n$ simulations?

* The analysis seems to hinge on rotational invariance, reducing everything to the scalar overlap $𝑚=⟨𝑤,𝑤⋆⟩$ and on the conditional expectation operator ​ being diagonal in the Hermite basis (so modes decouple and the dynamics close on $(a_t,m_t)$. Could you elaborate on which part of the proof fails for a multi-index target $x\mapsto \phi( (W^{*})^{\top}x)$?  In that case, one can try to prove the results of [1] ?

* Your main results are in the population and continuous-time regime. Do you expect the symmetry-breaking/escape mechanism to persist under SGD noise and finite truncation, and can you state a conjectured sample/step-size scaling? (Even a heuristic discussion would strengthen the “practical” claims.)


[1] "On learning Gaussian multi-index models with gradient flow part I: General properties and two-timescale learning;" Bietti, Alberto and Bruna, Joan and Pillaud-Vivien, Loucas

**Limitations:**

yes

**Strengths And Weaknesses:**

## Strengths
* The paper is well-written, clear, and easy to read. The Hermite reduction gives a clean, interpretable dynamical system for joint training in a nonconvex setting
* The paper sharply contrasts the planted flow (which can get stuck for $m_0<0$) with joint learning (which provably escapes this obstruction). The analysis suceeds to track the joint training, which is known to be hard (only [1] succeeds to do this, to my knowledge).
* The fast-slow mechanism and its dependence on the information exponent $s$ are well articulated.

## Weakness
* The qualitative “two-timescale / information exponent controls escape” theme overlaps with prior gradient-flow/SGD analyses in related Gaussian models; the paper would benefit from a sharper statement of what is genuinely new because $f$ is learned jointly rather than fixed. Besides, to my understanding, this idea has been used in the multi-index in order to make the landscape benign [2].

[1]"Sgd finds then tunes features in two-layer neural networks with near-optimal sample complexity: A case study in the xor problem." Glasgow, Margalit.

[2]"On learning Gaussian multi-index models with gradient flow part I: General properties and two-timescale learning;" Bietti, Alberto and Bruna, Joan and Pillaud-Vivien, Loucas

---

> ### Author Rebuttal · Authors · 2026-03-25
>
> We thank Reviewer tJw7 for their expert reading and supportive assessment. We address each point below.
>
> **On the overlap with prior two-timescale analyses and what is genuinely new.** The reviewer is right that the two-timescale / information exponent theme appears in prior work. Let us clarify what is genuinely new.
>
> Compared to [2] (BBP-V): in [2], the two-timescale structure is *designed a priori* — the algorithm alternates between updating the projection and updating the function in separate stages. In our work, the learning of the nonparametric function and the projection is **simultaneous**: both evolve under the same gradient flow, and the fast-slow separation is an *intrinsic property of the coupled equations*, not an algorithmic design choice. Analyzing this is substantially harder, because one must control the feedback loop between $a_{k,t}$ and $m_t$ without the luxury of freezing one while updating the other. This is a stronger result, though we note that [2] tackles a more challenging multi-index model.
>
> Compared to [1] (Glasgow): their analysis is specifically designed for the XOR model. The strength of their work lies in tracking joint training when the nonparametric function is learned via a *neural network*. Our framework uses an RKHS, which is simpler, but applies to a much broader class of target functions $\varphi^\star \in \mathcal{H}^1$.
>
> We will sharpen this comparison in the revised manuscript.
>
> **On the "variational principle".** We believe the reviewer is referring to what we call the *monotonicity principle* in Section 4.3: once sufficiently many coefficients $a_{k,t}$ have adopted the sign of $a_k^\star m_t^k$, the correlation $m_t$ becomes monotone. We will ensure the terminology is unambiguous in the revised version.
>
> **On well-definedness of the gradient flow in $L^2_\gamma$.** The system (8)–(9) on the reduced coordinates $(a_{k,t}, m_t) \in \ell^2(\mathbb{N}) \times [-1,1]$ is an ODE in a Hilbert space. The vector field is locally Lipschitz under the assumption $\varphi^\star \in \mathcal{H}^1$: the right-hand side of (8) is affine in $a_{k,t}$, and the series in (9) converges absolutely thanks to $\sum_k k^2 |a_k^\star|^2 < \infty$. A standard Cauchy–Lipschitz argument yields local existence and uniqueness; global existence follows from the a priori bounds $|m_t| \leq 1$ and $\|a_t\|_{\ell^2} \leq C$ (which follow from the Lyapunov structure of the loss). We will add this justification.
>
> **On two-timescale expansion language.** The timescale separation is an observation we make and exploit by tracking the system directly. We do not resort to general singular perturbation theory and do not know whether applying such a framework would simplify the argument — the non-standard features of our setting may make it more cumbersome than helpful. We will mention this connection in the revised version.
>
> **On multiple initializations.** The reviewer raises a valid practical point: one can run $n$ independent trials and keep the best. The planted failure is thus easily circumvented in practice. The interest of joint learning is primarily *theoretical and mechanistic*: it reveals that the flexibility of learning $\varphi$ eliminates spurious basins *dynamically*, without restarting. This is relevant for understanding why overparameterized models that learn both layers succeed where simpler models with fixed features fail — a phenomenon observed empirically but rarely explained rigorously.
>
> **On multi-index: which part of the proof fails.** Two ingredients break down for $x \mapsto \varphi^\star((W^\star)^\top x)$ with $W^\star \in \mathbb{R}^{d \times r}$, $r \geq 2$:
>
> 1. *The overlap is no longer scalar.* One must track the matrix $M_t = W_t^\top W^\star \in \mathbb{R}^{r \times r}$, and the conditional expectation operator $T_M$ is no longer diagonal in a simple basis — it involves multivariate Hermite polynomials, and the modes do not decouple.
>
> 2. *The monotonicity principle fails as stated.* In the single-index case, the sign of $\dot{m}_t$ is determined by a single power series. In the multi-index case, the dynamics of $M_t$ involve a matrix-valued vector field, and monotonicity has no direct scalar analogue.
>
> Extending the results to the multi-index joint learning setting, building on the planted analysis of [2], is a natural and important open problem.
>
> **On conjectured sample/step-size scaling.** We provide a detailed explanation of this in the answer of reviewer HV7h (see **Weakness 2 / Question 4**) and refer to it. We stress here rapidly that the reason of the difficulty lies in a rigorous treatment of the interaction between SGD noise and the fast-slow structure — during the slow phase, the signal-to-noise ratio in is very low, and finite-sample noise could interfere with the monotonicity principle. Our heuristic is that this interaction is in fact *benign*, for the following reason: the fast-slow decoupling naturally tempers the stochastic fluctuations at each phase.

---

> > ### Author Rebuttal · Reviewer_tJw7 · 2026-04-01
> >
> > Thank you to the authors for your thoughtful responses. I am satisfied with your answers.

---

### Official Review · Reviewer_HV7h · 2026-03-08

**Soundness:** 4
**Presentation:** 3
**Significance:** 3
**Originality:** 3
**Overall Recommendation:** 5
**Confidence:** 3

**Summary:**

This paper studies the joint learning of a hidden projection direction and a nonlinear link function in high-dimensional Gaussian single-index models. The authors analyze the gradient-flow dynamics of a learning procedure that simultaneously optimizes over both the projection direction and the function. Using a Hermite expansion, they characterize the dynamics in a reduced parameter space and establish convergence guarantees. A key consequence states that joint optimization can recover the target predictor even when the initial direction is negatively correlated with the true signal, implying potential benefits of joint learning. This contrasts with the planted model setting, where the link function is fixed and only the direction is optimized, in which case the dynamics may fail to recover the signal under the same conditions. Finally, the authors propose an implementation using a Hermite-based reproducing kernel Hilbert space (RKHS) and present numerical experiments that illustrate the theoretically predicted learning behavior.

**Compliance With Llm Reviewing Policy:**

Affirmed.

**Ethical Review Concerns:**

/

**Final Justification:**

I believe my concerns have been adequately addressed by the rebuttal. Accordingly, I have raised my score and recommend this work for acceptance.

**Key Questions For Authors:**

**Questions**
1. Is it obvious that the system defined in Eqs. (5)–(6), and equivalently in Eqs. (8)–(9), admits a unique and well-defined solution for all $ t $? In other words, is the gradient flow guaranteed to be uniquely defined for all times?
2. On page 3, what is meant precisely by “for $ g $ sufficiently smooth”? It would be helpful to specify the exact regularity assumptions on $ g $.
3. Several arguments rely on choosing a sufficiently large $ k^* $ so that certain tail sums are arbitrarily small. Is it clear that such a choice can be made uniformly over the entire trajectory of the dynamics, i.e., uniformly in time $ t $?
4. What additional ingredients would be required to extend the analysis from the population setting to a finite-sample setting?
5. What are the practical implications of the advantage of joint learning? In particular, the failure of the planted model under negative correlation follows from the fact that the function is fixed. However, if one allowed both $ \varphi^* $ and $ -\varphi^* $ in the planted model, the advantage disappears. Is there a known phenomenon or broader interpretation related to this effect?
6. In the experiments for Figure 1, why are the initializations chosen in the specific way described rather than sampling everything uniformly at random, as in the theoretical setup? This is mainly a personal curiosity.

**Typos encountered:**
- Page 2: The nested structure of functional spaces should be in reverse order of inclusions, as $L_\gamma^2$ is the largest.
- Page 6: In proposition 5.1. I suppose that $\beta$ should be $k$.
- Page 6: a Euler system -> an Euler system.
- Page 7: "showing that at this stage that last a time of order $d^{s-1}$" -> ....
- Page 7: the dynamics reaches -> the dynamics reach/the dynamic reaches
- Page 17: close forme -> close form

**Comment**
- On pages 12 and 13, the symbol $\Phi(x)$ is used in two different meanings. Although the intended meaning can be inferred from context, it could improve clarity to use different notation for the function $\Phi$ introduced in Section 4.1.

**Limitations:**

yes

**Strengths And Weaknesses:**

**Strenghts**
1. The paper is clearly written and generally easy to follow.
2. The use of the Hermite expansion yields a nice framework that reduces the infinite-dimensional problem to a tractable system.
3. The result showing that joint learning of the projection direction and the link function can outperform learning the direction when the link function is fixed, i.e. in the planted model, is interesting and somewhat non-obvious.

**Weaknesses**
1. Some minor formal details appear to be missing in the theoretical analysis.
2. The analysis is conducted in the population (infinite-sample) regime, which somewhat limits the impact for a theory paper without a clearer discussion of the finite-sample setting.
3. The connection to neural networks mentioned in the introduction is not clearly established in the main results, and currently feels somewhat speculative.
4. While the theoretical advantage of joint learning is interesting, the paper could better clarify the practical implications of this advantage.

---

> ### Author Rebuttal · Authors · 2026-03-26
>
> We thank Reviewer HV7h for their detailed reading and constructive questions. We address each weakness and question below.
>
> **Weakness 1 / Questions 1–3: Missing formal details.**
>
> We agree that these points deserve explicit treatment and will add them to the revised manuscript.
>
> *Q1 (Well-definedness and uniqueness of the flow).* Yes. The reduced system (8)–(9) is an ODE on $\ell^2(\mathbb{N}) \times [-1,1]$. The right-hand side of (8) is affine in $a_{k,t}$, and the series in (9) converges absolutely under the assumption $\varphi^\star \in \mathcal{H}^1$. The vector field is therefore locally Lipschitz, and Cauchy–Lipschitz yields local existence and uniqueness. Global existence follows from the bound $|a_{k,t}| \leq \max(|a_k^\star|, |a_{k,0}|)$ (Fact A.3 in the appendix) together with $|m_t| \leq 1$, which prevents finite-time blow-up.
>
> *Q2 ("For $g$ sufficiently smooth").* This phrasing is unnecessarily restrictive. Since the loss $\mathcal{L}(f, w)$ is quadratic in $f$, the Fréchet differential exists for all $g \in L^2_\gamma$ — no additional smoothness is needed. We will correct this.
>
> *Q3 (Uniform-in-time choice of $k^\star$).* Yes. The key observation is that $|a_{k,t}| \leq \max(|a_k^\star|, |a_{k,0}|)$ for all $t \geq 0$ (Fact A.3). Therefore the tail satisfies
>
> $\sum_{k \geq k^\star} k  |a_{k,t} a_k^\star| |m_t|^{k-1} \leq \sum_{k \geq k^\star} k \max(|a_k^\star|, |a_{k,0}|) |a_k^\star|,$
>
> which is the tail of a fixed convergent series ($\varphi^\star \in \mathcal{H}^1$, $f_0 \in \mathcal{H}^2$). The choice of $k^\star$ depends only on the target and the initialization, not on the trajectory.
>
> **Weakness 2 / Question 4: Population regime and finite-sample extension.**
>
> We acknowledge that the population setting is a limitation, and controlling finite-sample noise is the main challenge we could not address in the present article.
>
> We discuss here the keys and difficulties to get a sample complexity bound. The crux is that the fast-slow decoupling naturally tempers the SGD fluctuations at each phase:
>
> - *During the fast phase (coefficients $a_{k,t}$ equilibrate).* The stochastic fluctuations in the $a_k$-dynamics are proportional to $m_t^k$, which is very small at initialization. The low signal of $m_t$ thus acts as a natural damper on the noise, allowing step sizes of order $O(1)$ without destabilizing the fast equilibration.
> - *During the slow phase ($m_t$ increases).* Once the monotonicity principle is established, the evolution of $m_t$ depends on $a_{k,t}$, whose magnitudes are themselves still small (of order $|a_k^\star| m_t^k$). The fluctuations in the $m$-dynamics are therefore also tempered, again permitting $O(1)$ step sizes.
>
> We hence conjecture that the sample complexity $n = \text{time-complexity} / \text{step-size} \sim d^{s-1}$, matching the threshold for the planted case. Notably, no step-size scheduling is needed: the self-tempering of fluctuations is intrinsic to the joint dynamics, which further validates the adaptivity of the joint learning approach. We will add this heuristic discussion to Section 7. Figure 2 already provides empirical evidence that the predicted dynamical structure persists under finite samples ($n = 10^5$, $d = 100$); we will add error bars and further experiments in the revision.
>
> **Weakness 3: Connection to neural networks.**
>
> We agree that the connection as stated is too loose: any direct claim would be far-fetched. The only structural parallel is that our model has two layers — the projection $w$ and the nonlinear function $f$ expanded in the Hermite basis — mirroring the two layers of a single-neuron network.
>
> **Weakness 4 / Question 5: Practical implications and the $\{\varphi^\star, -\varphi^\star\}$ observation.**
>
> The reviewer is mathematically correct: planting both $\varphi^\star(\cdot)$ and $\varphi^\star(-\cdot)$ and keeping the best resolves the sign ambiguity (we believe the reviewer wanted to say $\varphi^\star (- \cdot)$ and not $-\varphi^\star (\cdot)$ --this latter does not enable recovery). More generally, as also noted by Reviewer tJw7, one can run multiple initializations and select the best. The advantage of joint learning over the planted model is not meant as a practical prescription — it is a *theoretical toy model* that provides mechanistic interpretability: the flexibility of learning $f$ simultaneously creates a self-correcting feedback loop that eliminates spurious basins dynamically. This gives a rigorous illustration of why learning both layers can help, which we believe is of conceptual value. We will clarify this framing in the revision.
>
> **Question 6: Choice of initialization in experiments.**
>
> The initialization $a_{k,0} = \pm 1$ is chosen for visual clarity: it makes the phase transitions clearly visible. A fully random initialization (as in the theoretical setup) produces the same qualitative behavior.
>
> **Typos and notation.** We thank the reviewer for the careful list. All typos will be corrected.

---

> > ### Author Rebuttal · Reviewer_HV7h · 2026-04-02
> >
> > I thank the authors for their thorough rebuttal and believe my concerns have been adequately addressed. Accordingly, I have raised my score and recommend this work for acceptance.

---

### Official Review · Reviewer_Qz8K · 2026-03-09

**Soundness:** 3
**Presentation:** 3
**Significance:** 3
**Originality:** 3
**Overall Recommendation:** 5
**Confidence:** 4

**Summary:**

The authors consider the problem of learning a single index model ($\varphi(w_\star^\top x)$) where both $w_\star$ and $\varphi$ are learnt jointly. The authors assume the input data to be gaussian as $x \sim N(0, I_d)$, the link function is a measurable function, $w_\star$ is a unit norm vector and the output is corrupted by a noise with finite variance. They consider the problem in the high dimensional limit.
The authors study the gradient flow dynamics where the function is updated to stay at all timesteps in L^2 and the vector $w_\star$ to always be on the sphere. This update translates into a system of ODE for the parameter $m = <w_t, w_\star>$ and the coefficients in the Hermite basis of the learnt link function.
After a presentation of what happens in the case where the link function is not learnt the author state the main theorem (4.2 in page 5) where they analyse the time for weak learnability of the target. The authors consider also the approximation with a finite dimensional RKHS.
Lastly the authors show some simulations.

**Compliance With Llm Reviewing Policy:**

Affirmed.

**Final Justification:**

The paper is a solid theoretical contribution analyzing joint learning in single-index models. The rebuttal fully addressed my concerns regarding figures, reproducibility, and the\$mathcal{H}^1$ assumption. I maintain my score in favor of acceptance.

**Key Questions For Authors:**

- Are the simulations shown in Section 6 simulations of the system of equations (8) and (9) or actual numerical experiments on finite dimensional data? How would the finite dimensional simulation compare to the theoretical curves?
- Can the authors expand a bit on the main text why one requires that $\varphi^\star \in \mathcal{H}^1$ and not have some other assumptions like $\varphi^\star \in L^2_\gamma$?

**Limitations:**

The authors have properly discussed limitations of the setup. Additionally to what the author already discussed it would be helpful for a reader to discuss wether the main behaviour is expected to hold also for more complex cases (such as multi-index) or for other data distributions.

**Strengths And Weaknesses:**

__Strengths__
- _Soundness_ The proof is rigorous and based on a series of fundamental works/techniques to study the weak learnability threshold of simple neural networks. What are some additional interesting points of the proof that where difficult to overcome with respect to previous works?
- _Presentation_ The work is well written and is able to convey the different results and their importance even without going into the technical details of the proof.
- _Significance&Originality_ The problem is a natural extension of the problems of weak learnability considered already in the literature. Its findings are interesting as it capture the interplay between link function and weights and shows the fact that this more generalised approach of learning them jointly can achieve better results.

__Weaknesses__
- _Presentation_ The paper would benefit by an additional discussion of related work on non-convex single-index and multi-index models. Given the current interest in non-convex estimation, it would be useful to position the contribution relative to prior work studying non-convex models with different techniques, especially regarding the characterisation of the optimisation landscape and its statistical implications.
Specifically Loh and Wainwright "High-dimensional regression with noisy and missing data: Provable guarantees with non-convexity" propose and study non-convex estimators for high-dimensional sparse linear regression, in Vilucchio et al. "Asymptotics of non-convex generalized linear models in high-dimensions: A proof of the replica formula" the authors study the landscape of generalised linear models with non-convex loss and regularisation functions and Montanari and Saeed "Topological trivialization in non-convex empirical risk minimization" consider the more challenging problem of multi-index.
When looking at Figure 3 (left panel) what is the reason to compare the parametric curve with $x = -y^4$? What should I learn from this figure?
An additional but minor point about the presentation is the fact that no informations are given on how to reproduce Figures 2 and 3.

---

> ### Author Rebuttal · Authors · 2026-03-25
>
> We thank Reviewer Qz8K for their careful reading and positive assessment. We address each point below.
>
> **On additional related work (non-convex landscape analysis).** The reviewer raises an important point. We will expand the discussion of related work, with a focus on the specific type of non-convexity at play: the *simultaneous* learning of both a projection direction and a function. Not all non-convex problems are equivalent, and ours is distinguished by this joint structure.
>
> - *Loh & Wainwright (2012):* They study non-convex (EM-like) procedures for an underlying *linear* problem under sparsity. The non-convexity is algorithmic, not structural in the sense of joint representation learning — there is no analogue of learning $\varphi$ and $w$ simultaneously.
> - *Vilucchio et al. (2024):* They characterize the asymptotic landscape of generalized linear models with non-convex penalties via the replica method, but in the planted/GLM regime where the link function is *known*.
> - *Montanari & Saeed (2024):* They provide DMFT equations and study topological trivialization of the landscape, but in the single-index (not multi-index) setting, and from a static landscape perspective rather than a dynamical one.
>
> We will add references to works where both features and projections are learned jointly, which is the relevant class of non-convex problems for our contribution.
>
> **On Figure 1 (left panel) and the curve $x = -y^4$.** The curve $a_{4,t} \approx a_4^\star m_t^4$ is the quasi-static equilibrium predicted by the fast-slow analysis (Phase I, Section 4.3). The left panel confirms that the trajectory tracks this algebraic curve as long as $m_t \ll 1$, and departs from it at the onset of Phase III. We will clarify this in the caption and add arrows indicating the direction of time.
>
> **Figures 1 and 2.** Figure 1 simulates the idealized ODE system (Eqs. (8)–(9)) via Euler discretization. Figure 2 implements the empirical RKHS-based gradient descent on finite-dimensional data ($n = 10^5$, $d = 100$). We will make this distinction more explicit in the revised text. Note that there is no Figure 3. We will also include full experimental details (step sizes, truncation level $\mathsf{k}$, initialization) and add error bars to Figure 2. We will also provide code. Thank you for pointing this out.
>
> **On technical novelty relative to prior work.** The main difficulty that distinguishes our analysis from the planted case is *the monotonicity principle.* One must show that sufficiently many coefficients $a_{k,t}$ adopt the correct sign (Lemma A.8) *before* $m_t$ has moved appreciably, and that the tail $\sum_{k \geq k^\star} k |a_{k,t} a_k^\star m_t^{k-1}|$ remains negligible uniformly in time. This fast-slow interplay is the core of the coupled analysis and relies on $\varphi^\star \in \mathcal{H}^1$.
>
> **Why $\varphi^\star \in \mathcal{H}^1$ rather than $\varphi^\star \in L^2_\gamma$.** The assumption $\sum_k k^2 |a_k^\star|^2 < \infty $ is used at a single but critical point: ensuring that the tail of the series in $\frac{d}{dt} m_t$, $\sum_{k \geq k^\star} k a_{k,t} a_k^\star m_t^{k-1},$ can be made arbitrarily small uniformly in time by choosing $k^\star$ large. Under the weaker $L^2_\gamma$ assumption, the factor $k$ is not compensated and the monotonicity principle breaks down. We believe the assumption could be relaxed (e.g., to $\mathcal{H}^{1/2+\varepsilon}$) at the cost of a more involved argument, which we leave for future work.
>
> **On multi-index and non-Gaussian extensions.** For multi-index models ($r \geq 2$), the Hermite expansion no longer diagonalizes the conditional expectation operator, and the dynamics do not reduce to a scalar overlap. The fast-slow mechanism and the advantage of joint learning are expected to persist qualitatively (as suggested by Bietti, Bruna & Pillaud-Vivien, 2023), but a rigorous treatment is an important open problem. For non-Gaussian data, the Hermite basis is specific to the Gaussian measure, but the broader phenomenon — joint learning escaping traps that the planted model cannot — should hold whenever the spectral structure of the conditional expectation operator is well-behaved, e.g., under a Poincaré inequality. We will expand the discussion in Section 7.

---

> > ### Author Rebuttal · Reviewer_Qz8K · 2026-04-01
> >
> > I thank the authors for the reply and for explaining carefully the point.
> > I think that with the improvements of the figures and captions the paper becomes even stronger. I additionally do not think that relaxing the assumption of $\varphi^\star \in \mathcal{H}^1$ is needed for this contribution as it is interesting in itself.
> > Sorry for referring to Figure 3, I was confused of the fact that Figure 1 comes after Figure 2, tis could be fixed for the camera ready version.
> > I maintain my score in favour of acceptance.

---

### Decision · Program_Chairs · 2026-04-30

**Decision:**

Accept (spotlight)

**Comment:**

This submission investigates learning the hidden projection direction and nonlinear link function in high-dimensional Gaussian single-index models through an analysis of its gradient-flow dynamics. All reviewers support acceptance. Clear accept.